# Mechanistic Insights into Grokking from the Embedding Layer

## Abstract

Grokking, a delayed generalization in neural networks after perfect training performance, has been observed in Transformers and MLPs, but the components driving it remain underexplored. We show that embeddings are central to grokking: introducing them into MLPs induces delayed generalization in modular arithmetic tasks, whereas MLPs without embeddings can generalize immediately. Our analysis identifies two key mechanisms: (1) Embedding update dynamics, where rare tokens stagnate due to sparse gradient updates and weight decay, and (2) Bilinear coupling, where the interaction between embeddings and downstream weights introduces saddle points and increases sensitivity to initialization. To confirm these mechanisms, we investigate frequency-aware sampling, which balances token updates by minimizing gradient variance, and embedding-specific learning rates, derived from the asymmetric curvature of the bilinear loss landscape. We prove that an adaptive learning rate ratio, $\frac{\eta_E}{\eta_W} \propto \frac{\sigma_{\max}(E)}{\sigma_{\max}(W)} \cdot \frac{f_W}{f_E}$, mitigates bilinear coupling effects, accelerating convergence. Our methods not only improve grokking dynamics but also extend to broader challenges in Transformer optimization, where bilinear interactions hinder efficient training.

## 1 Introduction

The phenomenon of grokking, in which a neural network exhibits delayed generalization after achieving close to or perfect training performance, has emerged as a compelling topic in deep learning. Initially observed in Transformer architectures by [19], grokking presents a puzzling challenge where models that seem to overfit to training data eventually demonstrate remarkable generalization capabilities after extensive training. Subsequent research has identified this phenomenon across various architectures, including convolutional neural networks (CNNs) and multi-layer perceptrons (MLPs) [13, 12]. Despite growing interest, the underlying mechanisms of grokking remain elusive.

Existing studies have sought to unravel grokking by exploring its connection to delayed robustness, local complexity, and model architecture [3, 6]. For instance, [6] suggest that grokking coincides with a phase transition in the linear regions of a model's input space, leading to robust partitions that enable generalization after extended training. Others have attributed grokking to emergent circuit behaviors or optimization dynamics [17, 21]. However, these studies often focus on high-level phenomena, overlooking the role of specific components, such as embedding layers, in shaping the dynamics of grokking.

In this work, we argue that embedding layers are central to understanding the grokking phenomenon. By introducing embedding layers into MLP architectures, we observe clear grokking patterns even in simple modular arithmetic tasks, such as modular addition. Interestingly, MLPs without embedding layers can often generalize without grokking, suggesting that embeddings introduce unique dynamics that delay generalization. Our analysis identifies two critical factors that influence these dynamics:

Submitted to 39th Conference on Neural Information Processing Systems (NeurIPS 2025). Do not distribute.

1. **Embedding update dynamics:** Embedding parameters are updated through gradient descent and weight decay. However, embeddings corresponding to tokens not present in a given batch are updated solely via weight decay or residual effects from previous gradients in optimizers like Adam. This imbalance delays stabilization and can hinder training, particularly for low-probability tokens.

2. **Coupling with the first-layer weights:** When embeddings are multiplied with the weights of the first layer, they form a bilinear interaction. This coupling introduces structural complexity into the optimization landscape, making the process more susceptible to saddle points and increasing the sensitivity to initialization.

Building on these insights, we propose two strategies to address and prove the hypotheses introduced for embedding layers. **First**: A refined sampling methodology that ensures more uniform updates across all embeddings, mitigating frequency imbalance. **Second**: A learning rate adjustment for embeddings, setting it higher than that of the rest of the model. This adjustment counteracts the coupling effect with the first-layer weights, enabling faster stabilization and reducing the risk of optimization stagnation. Our experiments demonstrate that these strategies not only accelerate the grokking process but also enable generalization in scenarios where traditional approaches fail.

Additionally, the bilinear coupling observed in embedding-based MLPs highlights broader challenges in optimizing Transformer architectures. Transformers, which rely on multiplicative interactions in attention mechanisms, exhibit similar issues due to the bilinearity of query, key, and value projections. While softmax attention and scaling by the dimensionality $d$ help smooth the optimization landscape, these mechanisms may still struggle with increased saddle points in certain layers [5]. In summary, this work contributes to the understanding of grokking and its broader implications for deep learning by:

- Highlighting the unique role of embedding layers in delaying generalization and their coupling with the first layer in MLPs.
- Proposing strategies to accelerate grokking, including refined sampling and embedding-specific learning rates.
- Connecting the challenges in embedding-based optimization to broader issues in Transformer training, such as bilinearity, saddle points, and the effectiveness of adaptive optimizers like Adam.

By bridging insights from grokking and Transformer optimization, we provide a unified perspective on the interplay between embedding dynamics, optimization challenges, and generalization.

## 2 Related Work

The phenomenon of grokking, where generalization emerges abruptly after prolonged overfitting, was first observed in transformers [19] and later extended to CNNs and ResNets [13, 12], indicating it is architecture-agnostic. Various explanations have been proposed. [7] attribute it to phase transitions in local complexity ("delayed robustness"), while others link it to circuit efficiency [17, 21, 11]. Though insightful, these perspectives don't fully explain the delayed generalization. Connections to double descent have also been explored [1, 16], but grokking's dynamics remain distinct.

The closest work to ours studies modular addition using permutation-equivariant models [15], where one-hot inputs interact with the first layer as a fixed embedding. Their analysis, however, is limited to modular tasks and specific activations. In contrast, we generalize across datasets and highlight how embedding layers, especially when trainable, interact bilinearly with downstream weights, affecting optimization dynamics.

Related studies like Tensor Programs IV [24] prescribe per-layer scaling based on width, assuming independent layer evolution. Our setup differs: the embedding layer's updates depend on both its own width and the spectrum of the coupled layer. Prieto et al. [20] connect delayed generalization to numerical instability (Softmax Collapse), proposing solutions that complement our focus on structural coupling and gradient imbalance.

Unlike works that focus on final representations [4], we analyze the embedding layer's evolving role during training. Even with one-hot inputs, its interaction with the first linear layer forms a learnable

embedding mechanism. Concurrent work shows that transferring embeddings from small to large models can accelerate grokking [23]; while we share this motivation, we also observe in preliminary trials that transferring other MLP layers may offer similar benefits.

Finally, the bilinear coupling we analyze in MLPs parallels challenges in Transformer architectures, where attention mechanisms introduce similar multiplicative dynamics. Prior work highlights how adaptive optimizers like Adam outperform SGD due to gradient noise and curvature heterogeneity [25, 10, 26]. Our findings help bridge these perspectives by showing how embedding-layer coupling shapes optimization and generalization.

# 3 Preliminaries

## 3.1 Embedding Layers

The Transformer model [22] utilizes a self-attention mechanism to capture dependencies between tokens. In this framework, embeddings map input tokens to high-dimensional vectors, which are processed through attention layers. These embeddings help the model capture contextualized representations. In contrast, MLPs rely on fully connected layers without attention mechanisms. We investigate the role of embeddings in MLPs, specifically how they improve model generalization. The core contribution of this work is to examine the role of embedding layers in MLPs. These layers map discrete tokens to dense, high-dimensional vectors, enabling models to handle non-linear tasks like modular arithmetic. Even with one-hot inputs—as studied in theoretical settings [2, 15]—the first weight matrix effectively functions as a learned embedding. Thus, embeddings, whether explicit or implicit, play a central role in shaping model dynamics. While commonly associated with Transformers, we focus on MLPs as a simpler and more interpretable setting. MLPs avoid the added complexity of self-attention while still exhibiting phenomena like grokking. Importantly, the bilinear coupling between embeddings and downstream weights, central to our analysis, also arises in Transformers but is further complicated by attention. Studying MLPs allows us to isolate and understand this coupling in a clean, controlled environment.

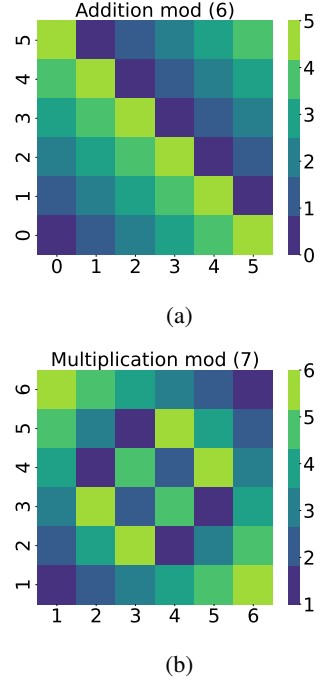

(a)

(b)

Figure 1: Heatmaps for (a) additive group (mod 6) and (b) multiplicative group (mod 7). The two groups are isomorphic despite differing appearances.

## 3.2 Algorithmic Datasets and Modular Arithmetic

Algorithmic datasets are synthetic datasets carefully constructed with controlled mathematical properties, typically involving operations over finite sets such as modular addition or multiplication. One well-known example is the modular arithmetic dataset studied by [19], where the goal is to uncover relationships between binary inputs and produce consistent outputs based on these operations. For instance, given inputs a and b, the model is tasked to compute $(a + b) \bmod P$ or $(a \times b) \bmod P$, where $P$ is a prime number, and both inputs and outputs are constrained within $\{0, 1, \ldots, P - 1\}$ (refer to Figure 1).

This dataset highlights the challenging nature of generalization in grokking: the relationship between inputs is defined purely by a deterministic operation, not by a probabilistic distribution. Unlike typical machine learning datasets, where examples are drawn from an underlying (often unknown) data distribution, algorithmic datasets consist of a finite and complete set of all possible input-output combinations. In such cases, there is no statistical "distribution" in the conventional sense; instead, the generalization task relies on uncovering the underlying relationship between inputs, which demands a model to internalize the algorithm itself. Moreover, any hypothesis consistent with training examples can initially seem plausible from a statistical perspective, as no known distribution governs the data.

The difficulty of generalization thus lies not in interpolating unseen samples but in discovering the underlying relation, making it a fundamentally different task.

We note that there is an equivalence between modular addition and modular multiplication in certain settings. Namely, given a prime number $p$, the groups (in mathematical sense) of modular addition $\big(\{0, 1, \ldots, p-2\}, +\big)$ (where addition is performed modulo $p-1$), and of modular multiplication $\big(\{1, \ldots, p-1\}, *\big)$ (where multiplication is performed modulo $p$) are isomorphic. Both groups have the same number of elements (which is $p-1$), and are simple (meaning, there is an element $g$, called generator, such that every other element is of the form $g * \cdots * g$, where $*$ is the group operation and the number of operations used is less than $p$. In the first group, any element different from $0$ is the group generator while in the second group, any element different from $1$ is the generator (see Figure 1).

The embedding layer strips the input group elements of their numerical meanings, and assigns a general, abstract vector to each element. In this way, training on modular addition or multiplication presents no difference for MLP (or other architectures) with the embedding layer. In contrast to this, the MLP without the embedding layer is able to fit and generalize on modular addition, while it completely fails on modular multiplication.

### 3.3 Problem Setup and Motivations

Let $\mathcal{D} = \{(x_i, y_i)\}_{i=1}^{N}$ represent an algorithmic dataset, where each $x_i$ is an input token sequence (e.g., $a, b$, `operation`, `=`), and $y_i$ is the output derived from an operation modulo a positive integer $P$. The task is to learn a mapping $f_\theta : \mathcal{X} \to \mathcal{Y}$ parameterized by $\theta$, capable of generalizing to unseen samples from $\mathcal{D}_{\text{test}}$.

To process inputs effectively, we tokenize them as sequences of their digit representations, as the model does not inherently interpret numerical values. Each operand $a$ and $b$ is assigned a token in the range $0$ to $P-1$, while the operation and equality symbols are represented by tokens $P$ and $P+1$, respectively. For instance, the modular arithmetic expression $(3 + 2)(\bmod\ 5) = 0$ is tokenized as $[3, 5, 2, 6, 0]$.

Embedding layers in models provide a dense representation of tokens. However, delayed updates to embeddings for infrequent tokens can significantly impact convergence and generalization. Our work explores these dynamics, with a focus on the impact of $p_i$, the $i^{th}$-token sampling probability, and proposes adjustments to improve convergence. We investigate the use of embeddings in MLPs for algorithmic tasks. We started by training a MLP on modular addition and multiplication datasets, comparing setups with and without embedding layers.

**MLP Without Embeddings.** In this setup, input tokens ($a$, $b$, operation ($P$), and equality sign ($P+1$)) are encoded directly into a 4-dimensional input vector. The MLP processes these inputs as:

$$\boldsymbol{h}_1 = \sigma(\mathbf{W}_1 x + \boldsymbol{b}_1), \quad \boldsymbol{h}_2 = \mathbf{W}_2 \boldsymbol{h}_1 + \boldsymbol{b}_2,$$
$$\hat{\boldsymbol{y}} = \text{Softmax}(\boldsymbol{h}_2). \tag{1}$$

where $\boldsymbol{x} \in \mathbb{R}^4$ is the encoded input vector (with first and third entry $a$ and $b$, respectively), $\mathbf{W}_1, \mathbf{W}_2$ are weight matrices, $\boldsymbol{b}_1, \boldsymbol{b}_2$ are biases, $\sigma$ is the ReLU activation function, and $\hat{\boldsymbol{y}}$ represents the predicted output.

This configuration demonstrates that the MLP can fit the addition task with ease, but struggles to generalize multiplication. This difficulty arises because multiplication modulo $P$ is not linearly separable, as evident in the non-trivial patterns in Figure 1.

**MLP With Embeddings.** To overcome the challenges of non-linear separability, we introduced an embedding layer. Each token $x$ is mapped to a dense vector $\boldsymbol{e}_x$ through an embedding matrix $\mathbf{E} \in \mathbb{R}^{V \times d}$, where $d$ is the embedding dimension. Our input consists of 4 token embeddings of the form $\hat{\boldsymbol{e}} = [\boldsymbol{e}_i, \boldsymbol{e}_{'*'}, \boldsymbol{e}_k, \boldsymbol{e}_{'='}]^\top$, and the modified forward pass is:

$$\boldsymbol{h}_1 = \sigma(\mathbf{W}\hat{\boldsymbol{e}} + \boldsymbol{b}_1),$$
$$\boldsymbol{h}_2 = \mathbf{W}_2 \boldsymbol{h}_1 + \boldsymbol{b}_2, \quad \hat{\boldsymbol{y}} = \text{Softmax}(\boldsymbol{h}_2), \tag{2}$$

Adding embeddings allows the model to capture more expressive input representations. With this setup, we observed that the model generalized well to both addition and multiplication tasks, but with a delayed generalization for multiplication. This delay corresponds to the grokking phenomenon, which appears as a "trapezoid pattern" in performance plots: a phase of memorization followed by a sudden leap in test accuracy, as illustrated in figure 2 .

These observations motivate a deeper analysis of embedding dynamics during training. In particular, we investigated the gradient heatmaps to understand the role of embeddings in delaying generalization. By visualizing gradient magnitudes across training epochs, we point out that embeddings receive smaller updates compared to other weights of the model, potentially causing grokking. This investigation will help establish a connection between embedding behavior and the observed generalization delays.

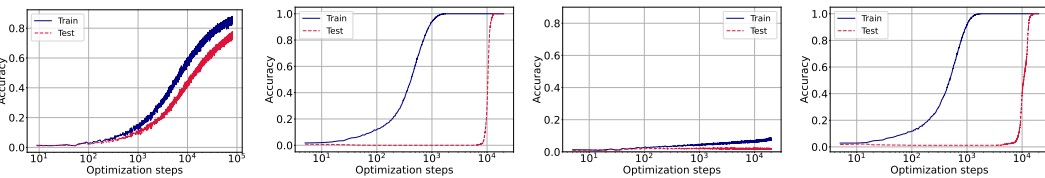

Figure 2: Training and validation accuracies of the MLP model on modular arithmetic tasks, trained with Adam. *Left two:* Addition task, without (first) and with (second) embeddings. *Right two:* Multiplication task, without (third) and with (fourth) embeddings. In the embedding-free cases, training and validation accuracies increase together only for addition; multiplication fails to generalize. In contrast, models with embeddings reach 100% training accuracy in both tasks, but only begin generalizing after a delay exhibiting the grokking phenomenon.

# 4 Main Results

Our methodology investigates the dynamics of embedding layers within MLPs to address challenges in generalization, particularly in the context of algorithmic tasks. The key contributions include: (1) exploring the novel role of embedding layers attached to MLP architectures, (2) examining the impact of embedding sampling probability $p_i$ on training dynamics, and (3) understanding how initialization and the coupling of embedding and weight matrices affect learning efficiency. These factors contribute to the grokking phenomenon, where generalization is delayed during training.

## 4.1 Embedding Dynamics

Let the loss function of the model be $\mathcal{L}(\theta, \mathbf{E})$, where $\theta$ is model parameters other than embedding weights. Let $e_{i,t}$ denote the embedding vector for token $i$ at step $t$. Under stochastic gradient descent (SGD) with weight decay $\lambda$, the embedding update rule is:

$$e_{i,t+1} - e_{i,t} = -\eta\lambda e_{i,t} - \eta\nabla_{e_{i,t}}\mathcal{L}, \tag{3}$$

where $\eta$ is the learning rate, and $\nabla_{e_i}\mathcal{L}$ is the gradient[1]. Token embeddings are updated using corresponding gradients only when the associated tokens appear in a batch. Assume that token $i$ being sampled in a batch with a probability $p_i$. Consequently, taking into account the randomness of batch sampling, the expected update can be expressed as:

$$\mathbb{E}[e_{i,t+1} - e_{i,t}] = -\eta\lambda e_{i,t} - \eta p_i\nabla_{e_{i,t}}\mathcal{L}. \tag{4}$$

To summarize, the sampling probability $p_i$ directly influences the gradient dynamics of the embedding layer. While gradients contribute to tokens only probabilistically, weight decay affects all embeddings uniformly, leading to imbalances in parameter updates. This dynamic, visualized in Figure 3, highlights the need for a deeper understanding of how $p_i$ affects convergence.

To analyze the reduction of the loss, we assume that the model's overall loss function $\mathcal{L}(\theta, \{e_i\})$ is $\beta$-smooth. This means it satisfies the following inequality for all updates:

$$\mathcal{L}(\theta_{t+1}, \{e_{i,t+1}\}) \leq \mathcal{L}(\theta_t, \{e_{i,t}\}) + \langle\nabla\mathcal{L}, \Delta\rangle + \frac{\beta}{2}\|\Delta\|^2.$$

---

[1]Assuming the SGD update rule without momentum.

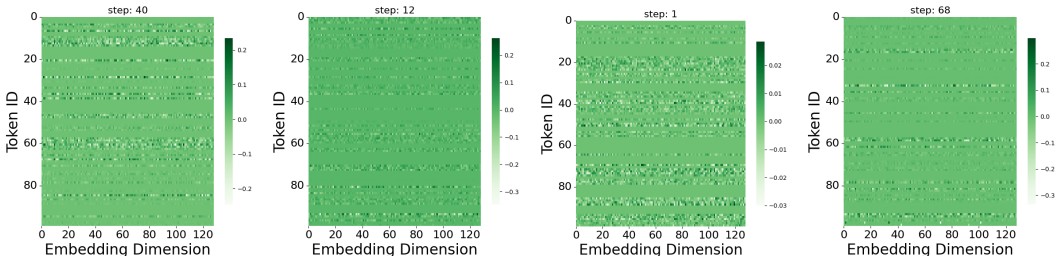

Figure 3: Gradient heat maps of the MLP model at random optimization steps. Sparse columns in the embedding gradients reflect the absence of certain tokens in sampled batches, leading to uneven learning dynamics and contributing to delayed generalization.

where $\Delta = (\theta_{t+1} - \theta_t, \boldsymbol{e}_{i,t+1} - \boldsymbol{e}_{i,t})$.

Denote $\mathcal{L}_t := \mathcal{L}(\theta_t, \{\boldsymbol{e}_{i,t}\})$ then taking expectations over randomness of batch sampling leads to the following expected update:

$$\mathbb{E}[\mathcal{L}_{t+1} - \mathcal{L}_t] \leq \nabla_{\theta_t}\mathcal{L}^T(\theta_{t+1} + \theta_t)$$
$$- \sum_{i=1}^{V} \nabla_{\boldsymbol{e}_{i,t}}\mathcal{L}^T \mathbb{E}(\boldsymbol{e}_{i,t+1} - \boldsymbol{e}_{i,t}) + \frac{\beta}{2}\|\Delta\|^2, \tag{5}$$

Substituting the embedding update based on equation 4 into the smoothness inequality,

$$\mathbb{E}[\mathcal{L}_{t+1} - \mathcal{L}_t] \leq \nabla_{\theta_t}\mathcal{L}^T(\theta_{t+1} - \theta_t)$$
$$- \eta \sum_{i=1}^{V} \left( p_i \|\nabla_{\boldsymbol{e}_{i,t}}\mathcal{L}\|^2 + \lambda \boldsymbol{e}_{i,t}^T \nabla_{\boldsymbol{e}_{i,t}}\mathcal{L} \right) + \frac{\beta}{2}\|\Delta\|^2, \tag{6}$$

and noting from the right hand side of the inequality above, $p_i$ plays important role in reduction of the expected loss. However, the dependence on $p_i$, is coupled with weight decay, which explains why these two parameters are important to study more deeply to draw a conclusion about grokking.

## 4.2 Dataset Splitting Strategies

To further explore the role of $p_i$, we investigate how train-test splitting strategies affect its value and, consequently, the grokking process. The train-test split determines the probability of token $i$ appearing in a batch.

We begin by assuming that the weight decay parameter $\lambda$ is zero and that the learning rate $\eta$ is uniform across all parameters. This reduces the optimization problem to focusing on $p_i$, under the constraints $\sum_{i=1}^{V} p_i = 1, p_i \geq 0 \,\forall i$. Specifically, the optimal $p_i$ can be found by solving for the following:

$$\min_{p_i | p_i \geq 0, \sum p_i = 1} -\eta \sum_{i=1}^{V} p_i \|\nabla_{\boldsymbol{e}_{i,t}}\mathcal{L}\|^2. \tag{7}$$

However, solving this exactly is challenging in practice due to the need for estimating all embedding gradient norms. Instead, we adopt approximate strategies for splitting the training data, guided by various assumptions about the gradient structure (see Appendix A for details).

1. **Uniform Sampling:** Distribute all combinations of $a$ and $b$ evenly across training and test sets.

2. **Skewed Sampling:** Introduce a bias in the combinations of $a$ that are distributed across training and test sets.

3. **Random Sampling:** Randomly distribute the examples across training and test sets.

These splits enable us to regulate token sampling probabilities, offering a direct assessment of the impact of $p_i$ on embedding convergence and grokking. Furthermore, Section 5.1 provides a detailed experiments conducted on two algorithmic datasets.

### 4.3 Embedding Convergence and Initialization

While the frequency of embedding updates plays a crucial role in training dynamics, as demonstrated in our experiments, it alone cannot fully explain phenomena such as grokking after fitting, its relationship to initialization, weight decay, or the structure of the loss landscape.

Stabilization (or convergence) occurs when the embedding $\mathbf{e}_i$ reaches a steady state where the updates become negligibly small, i.e., when the change in the embedding $\|e_{i,t+1} - e_{i,t}\|$ is approximately zero. This condition implies that, $(\eta\lambda)\boldsymbol{e}_{i,t} \approx \eta p_i \nabla_{\mathbf{e}_i}\mathcal{L}$. from equation 4.

For small learning rates ($\eta \ll 1$), the embedding updates behave like a continuous system, and we can model this as a differential equation (along every dimension):

$$\frac{d\mathbf{e}_i}{dt} = -\lambda\mathbf{e}_i - p_i\nabla_{\mathbf{e}_i}\mathcal{L}, \qquad (8)$$

where $\nabla_{\mathbf{e}_i}\mathcal{L}$ is the gradient of the loss function with respect to the embedding $i$. Assuming that the gradient $\nabla_{\mathbf{e}_i}\mathcal{L}$ stabilizes to a constant value $g$, the solution to this equation is:

$$\mathbf{e}_i(t) = Ce^{-\lambda t} - \frac{\eta p g}{\lambda}, \qquad (9)$$

where $C$ is an integration constant determined by the initial conditions. As time $t$ increases, the embedding $\mathbf{e}_i(t)$ converges to the equilibrium value $\mathbf{e}_i(t) \to -\frac{\eta p g}{\lambda}$. Thus, convergence is achieved when $\mathbf{e}_i(t)$ stabilizes around this

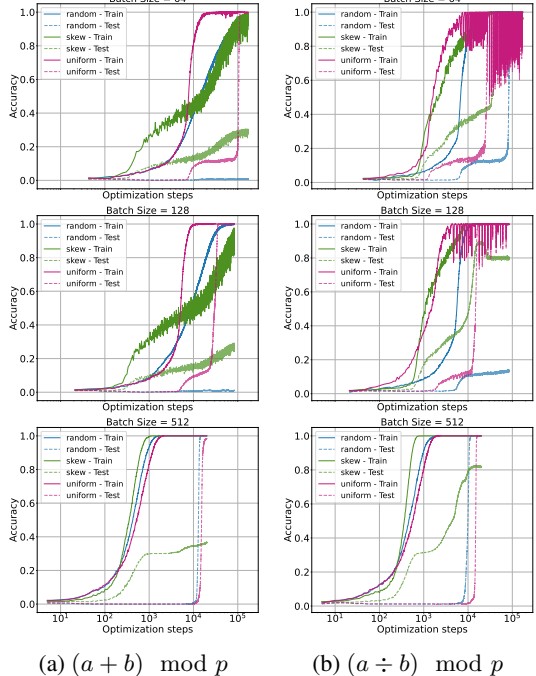

(a) $(a + b) \mod p$     (b) $(a \div b) \mod p$

Figure 4: Sampling strategy comparison for two modular tasks—addition and division—across all batch sizes. Uniform sampling generalizes faster; skewed sampling fails to generalize due to token imbalance.

equilibrium point. The time $T$ to reach convergence is bounded as $T \geq \frac{1}{\lambda}\ln\left(\frac{C}{\epsilon}\right)$, where $\epsilon$ is a small threshold. In summary, convergence time is governed by the embedding gradient $g$, the weight decay $\lambda$, and the initialization magnitude $C$: stronger gradients and larger $\lambda$ accelerate convergence, while larger initial values $C$ slow it down.

In bilinear models such as MLPs and Transformers, embedding gradients are tightly coupled with those of downstream weights (e.g., $\mathbf{W}$), forming a feedback loop: poor updates to $\mathbf{E}$ degrade $\mathbf{W}$, and vice versa. To study the role of initialization in this dynamic, we tested two setups: frozen embeddings, which led to slow convergence due to limited representational flexibility; and small initial embeddings, which improved convergence by allowing stronger early gradients—an effect also observed in prior work [26, 12], though without analyzing embedding-weight coupling.

Motivated by these observations, we propose the **Adam-LR Optimizer**, which adjusts the embedding learning rate to balance update magnitudes between $\mathbf{E}$ and $\mathbf{W}$. This coupling-aware scaling is formalized below:

**Proposition 4.1.** *Let $\mathbf{E}$ and $\mathbf{W}$ be the embedding matrix and first-layer weights. To equalize update scales under cross-entropy loss, the learning rate ratio $c = \frac{\eta_E}{\eta_W}$ should satisfy:*

$$c \propto \frac{\sigma_{\max}(\mathbf{E})}{\sigma_{\max}(\mathbf{W})} \cdot \frac{f_W}{f_E},$$

*where $\sigma_{\max}(\cdot)$ denotes the largest singular value and $f_E, f_W$ are the respective update frequencies,(see appendix B for details).*

In practice, we set $c = 10$, guided by empirical singular value trends and supported by sensitivity analysis (see Fig. 7, §5.2). This adjustment improves convergence and stability, especially under sparse embedding updates common in skewed token distributions.

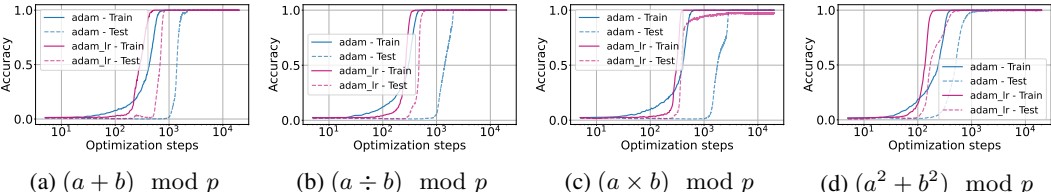

(a) $(a + b) \mod p$     (b) $(a \div b) \mod p$     (c) $(a \times b) \mod p$     (d) $(a^2 + b^2) \mod p$

Figure 5: Performance comparison of Adam-LR and Adam optimizers on four algorithmic datasets. Adam-LR scales the embedding learning rate based on the singular values of the embedding matrix. This adaptive adjustment accelerates convergence and enhances generalization across all datasets. The results demonstrate that Adam-LR significantly speeds up the grokking process compared to the standard Adam optimizer under identical training settings ($lr = 0.01$, batch size = 512).

## 5 Experiments and Discussions

We begin our exploration with a MLP model. The architecture consists of two layers, where the hidden dimension of the first layer is set to four times the embedding dimension (where four is the sequence length), and embedding dimension is set to 128, as per prior work on grokking. The second layer has a dimension of $P = 97$. The activation function used throughout is ReLU, and optimization is performed using the Adam optimizer with a weight decay of 0.001.

### 5.1 The Effect of Embedding Probability

The first set of experiments investigates various strategies for splitting the training and testing datasets. Specifically, we explore three approaches, namely; uniform sampling, skewed sampling, and random sampling.

The expression $(a + b) \mod p$ represents the sum of $a$ and $b$ modulo $p$. For our experiments, we randomly set aside 20% of the data as a test set, ensuring that evaluation is performed on unseen samples. From the remaining data, $30/80\%$ (i.e. 30% from total set) is sampled as the training set according to each sampling strategy.

Figure 4 compare the performance of the sampling methods (random, uniform, skew) across different splits of the dataset (see appendix D.1 for further datasets and settings). Each represents a specific datasets, while the rows compare batch sizes, and columns compare datasets. The x-axis is logarithmic to emphasize the convergence trends.

Uniform sampling generally promotes faster generalization and convergence compared to random sampling. However, its benefits diminish at larger batch sizes (e.g., beyond 512), where random sampling becomes nearly as effective due to broader token coverage. Crucially, our results show that skewed sampling—despite fitting the training data and preserving the overall train-test ratio—consistently leads to suboptimal generalization. This suggests that models can converge to lower subaccuracy plateaus when token probabilities are heavily imbalanced. Importantly, even uniform sampling does not guarantee optimality: unless the batch size is sufficiently large, some tokens may be consistently omitted from updates. These findings underscore that token probability, both in expectation and in per-batch coverage, plays a central role in embedding dynamics and grokking behavior.

### 5.2 Comparison of Optimizers

To evaluate the effectiveness of our proposed optimizer, Adam-LR, which incorporates a simple yet effective strategy for treating the embedding layer differently to avoid stagnation or saddle points, we conducted experiments on four datasets. The results are shown in Figure 5, where we compare the performance of the two optimizers, Adam-LR and the standard Adam optimizer, under identical training settings ($lr = 0.01$, batch size = 512).

Using our proposed optimizer, Adam-LR, which scales the embedding learning rate by a factor of 10, the results demonstrate a significant acceleration in the grokking process compared to the baseline Adam optimizer across all datasets.

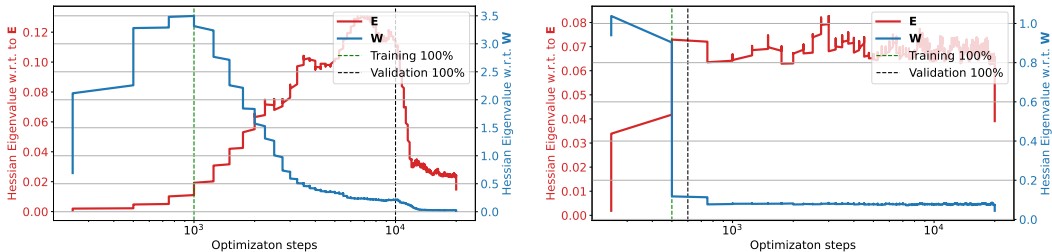

Figure 6: Maximum eigenvalues of the Hessian with respect to embedding weights (**E**) and down-stream weights (**W**) during training. The left plot corresponds to the Adam optimizer, while the right plot uses Adam_lr optimizer (ours). With Adam (left), the eigenvalues for **E** are significantly smaller than those for **W**, reflecting differences in dimensionality and update frequency. In contrast, with Adam_lr (right), the eigenvalues of **W** are notably reduced and become closer to those of **E**, suggesting a more balanced optimization dynamic. Training accuracy reaches 100% when the eigenvalues of **W** begin to decrease, while validation accuracy improves as the eigenvalues of **E** decrease. This suggests that **W** drives early optimization progress, while **E** fine-tunes generalization. The Adam_lr optimizer (ours) appears to regularize **W**, leading to a more stable training process.

## 5.3 Analysis of singular values of embedding layer

Prior work attributes Adam's superiority over SGD in Transformers to factors like gradient noise, descent direction, and Hessian block heterogeneity [25, 10, 18, 26]. However, these studies largely overlook the role of embeddings and their bilinear interactions. Our analysis supports the view that such bilinear structure, especially in embeddings, contributes significantly to the observed curvature differences (see appendix C.1 for more discussion).

To analyze the curvature of the loss landscape, we compute the maximum eigenvalue of the Hessian matrix using the power method with Hessian-vector products (HVPs). Figure 6 shows the maximum eigenvalues of the Hessian with respect to **E** and **W** during training. The results highlight distinct curvature properties for **E** and **W**, reflecting their roles in the bilinear interaction.

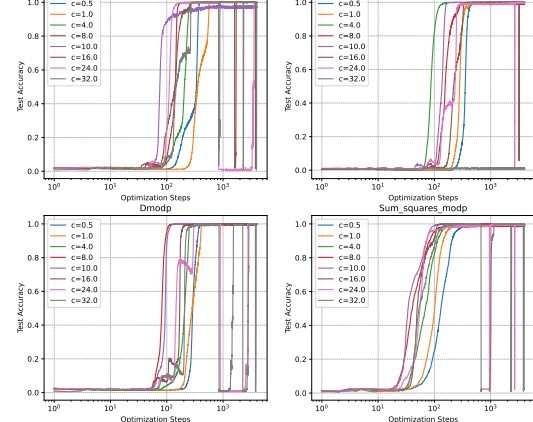

## 6 Discussions

In this study, we explored the interplay between embedding layers and downstream weights in neural networks, highlighting how their bilinear coupling influences optimization and drives the grokking phenomenon. We demonstrated that embedding layers play a central role in delayed generalization and introduced the Adam-LR optimizer to address the imbalance in update dynamics, scaling the embedding learning rate based on singular values and update frequencies.

A key limitation of this work is its focus on MLPs, which provide a simplified setting for analyzing embedding-weight coupling. While this enables controlled analysis, it leaves open how these insights transfer to more complex architectures such as Transformers, where similar

Figure 7: Sensitivity of test accuracy to the learning rate ratio $c = \eta_E / \eta_W$ across four tasks. Small $c$ leads to under-updating, large $c$ causes instability, and $c = 10$ consistently balances convergence and stability.

bilinear interactions appear in attention mechanisms but with added structural complexity. Extending our framework to the Transformer setting is a promising direction for future work.

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

 **Appendix**

 # A   Optimizing for Sampling Porbability

420 **Uniform Importance Assumption**

421 If we assume that all gradients are equally important, i.e., $\|\nabla_{\mathbf{E}_{i,t}}\mathcal{L}\|^2$ is uniform across all embed-
422 dings:

$$\|\nabla_{\mathbf{E}_{i,t}}\mathcal{L}\|^2 = c, \quad \forall i,$$

423 where $c$ is a constant.

424 In this case, the optimization of $-\sum_{i=1}^{V} p_i \|\nabla_{\mathbf{E}_{i,t}}\mathcal{L}\|^2$ becomes independent of $p_i$. To satisfy the
425 normalization constraint $\sum_{i=1}^{V} p_i = 1$, the optimal solution is:

$$p_i = \frac{1}{V}, \quad \forall i. \tag{10}$$

426 This corresponds to a uniform distribution, where all embeddings are treated equally (see Figure
427 8). While computationally efficient, this approach may lead to suboptimal convergence if some
428 embeddings contribute disproportionately to the loss reduction.

429 **Gradient Norm Bounded by $L_i$**

430 Now, let us assume that the gradient norm for each embedding is bounded,

$$\|\nabla_{\mathbf{E}_{i,t}}\mathcal{L}\| \leq L_i, \quad \forall i, \tag{11}$$

431 where $L_i$ is a known upper bound for embedding $i$. Using this bound, we approximate,

$$-\sum_{i=1}^{V} p_i \|\nabla_{\mathbf{E}_{i,t}}\mathcal{L}\|^2 \geq -\sum_{i=1}^{V} p_i L_i^2. \tag{12}$$

432 To maximize $\sum_{i=1}^{V} p_i L_i^2$ subject to the constraint $\sum_{i=1}^{V} p_i = 1$, we note that the objective function
433 is linear in $\mathbf{p}$. Therefore, the maximum is attained at a vertex of the probability simplex, meaning the
434 optimal solution is:

$$p_k = 1, \quad \text{where} \quad k = \arg\max_i L_i^2, \quad \text{and} \quad p_i = 0, \quad \forall i \neq k. \tag{13}$$

435 This result indicates that the optimal probability distribution assigns all weight to the embedding with
436 the highest gradient bound, ignoring all others. Therefore, to obtain a smooth probability distribution,
437 we introduce an entropy regularization term as follow,

$$H(\mathbf{p}) = -\sum_{i=1}^{V} p_i \log p_i. \tag{14}$$

438 We now optimize the modified objective,

$$\sum_{i=1}^{V} p_i L_i^2 + \gamma H(\mathbf{p}), \tag{15}$$

439 subject to the constraint $\sum_{i=1}^{V} p_i = 1$, where $\gamma > 0$ controls the strength of the regularization.

440 The corresponding Lagrangian is as follow,

$$\mathcal{L}_p = \sum_{i=1}^{V} p_i L_i^2 + \gamma \left( -\sum_{i=1}^{V} p_i \log p_i \right) + \mu \left( \sum_{i=1}^{V} p_i - 1 \right). \tag{16}$$

441 Taking the derivative with respect to $p_i$ and setting it to zero, we get,

$$L_i^2 - \gamma(1 + \log p_i) + \mu = 0. \tag{17}$$

Solving for $p_i$ gives:

$$\log p_i = \frac{L_i^2 + \mu - \gamma}{\gamma} \quad \Longrightarrow \quad p_i = \exp\left(\frac{L_i^2 + \mu - \gamma}{\gamma}\right). \tag{18}$$

Applying the constraint $\sum_{i=1}^{V} p_i = 1$, would results in the following solution,

$$p_i^* = \frac{\exp\left(L_i^2/\gamma\right)}{\sum_{j=1}^{V} \exp\left(L_j^2/\gamma\right)}. \tag{19}$$

This result smoothly distributes probabilities based on the gradient bounds, assigning higher probability to embeddings with larger $L_i^2$ while ensuring a non-degenerate distribution.

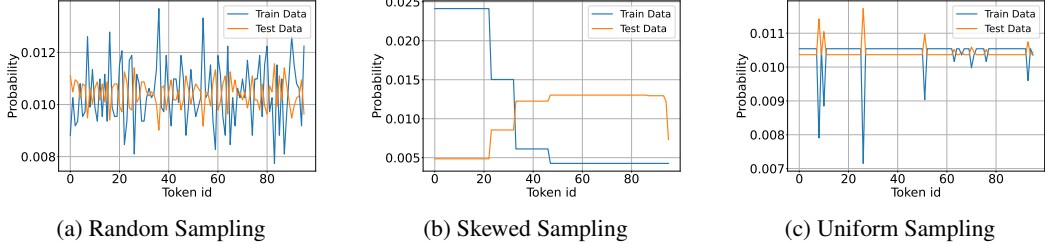

(a) Random Sampling      (b) Skewed Sampling      (c) Uniform Sampling

Figure 8: Token probabilities in the training and test sets under different sampling strategies. Imbalanced sampling leads to uneven token occurrences in mini-batches, causing some tokens to be absent in multiple updates while others appear frequently. This results in highly variable gradient updates, where frequently seen tokens converge faster, while rare tokens stagnate due to sparse updates, affecting overall model generalization.

# B    Dynamics of Updates in Bilinear Systems with Initialization Effects

We analyze the interaction between embeddings $\mathbf{E} \in \mathbb{R}^{p \times d}$ and weight matrix $\mathbf{W} \in \mathbb{R}^{4d \times d}$ in a bilinear term:

$$z(\mathbf{EW}), \tag{20}$$

where $z$ is an activation function applied elementwise. The gradients of $\mathbf{E}$ and $\mathbf{W}$ are given as:

$$\nabla_{\mathbf{E}} \propto \mathbf{W}^\top \nabla_{\text{loss}}, \quad \nabla_{\mathbf{W}} \propto \mathbf{E}^\top \nabla_{\text{loss}}. \tag{21}$$

The gradient norms are influenced by the dominant singular values of $\mathbf{W}$ and $\mathbf{E}$. Specifically:

$$\|\nabla_{\mathbf{E}}\| \propto \sigma_{\max}(\mathbf{W}), \quad \|\nabla_{\mathbf{W}}\| \propto \sigma_{\max}(\mathbf{E}). \tag{22}$$

At initialization, $\mathbf{E}$ and $\mathbf{W}$ are often drawn from distributions with variances that depend on their dimensions (e.g., PyTorch initializes weights with $\mathcal{N}(0, \sqrt{2/d})$ scaling). This initialization typically ensures $\sigma_{\max}(\mathbf{E}) \gg \sigma_{\max}(\mathbf{W})$, as $\mathbf{W}$ is higher-dimensional, amplifying the difference in gradient magnitudes.

The embedding matrix $\mathbf{E}$ is updated less frequently than $\mathbf{W}$ because not all tokens appear in every batch. Let $f_E$ and $f_W$ represent the update frequencies of $\mathbf{E}$ and $\mathbf{W}$, respectively. Typically, $f_W > f_E$, exacerbating the update disparity.

To balance the effective updates of $\mathbf{E}$ and $\mathbf{W}$, the learning rates $\eta_E$ and $\eta_W$ must be scaled to account for both their singular values and update frequencies. The effective update ratio is:

$$\frac{\|\Delta\mathbf{E}\|}{\|\Delta\mathbf{W}\|} \propto \frac{\eta_E \cdot \sigma_{\max}(\mathbf{W}) \cdot f_E}{\eta_W \cdot \sigma_{\max}(\mathbf{E}) \cdot f_W}. \tag{23}$$

For proportional updates ($\|\Delta\mathbf{E}\| \sim \|\Delta\mathbf{W}\|$), the ratio $c = \frac{\eta_E}{\eta_W}$ must satisfy:

$$c \propto \frac{\sigma_{\max}(\mathbf{E})}{\sigma_{\max}(\mathbf{W})} \cdot \frac{f_W}{f_E}. \tag{24}$$

The term $\frac{\sigma_{\max}(\mathbf{E})}{\sigma_{\max}(\mathbf{W})}$ reflects the imbalance in singular values due to initialization and structural properties. The term $\frac{f_W}{f_E}$ accounts for the frequency imbalance in updates between $\mathbf{E}$ and $\mathbf{W}$, driven by sparse token appearances in batches.

PyTorch initialization, which scales weights by $\mathcal{O}(\sqrt{2/d})$, ensures that $\sigma_{\max}(\mathbf{W})$ and $\sigma_{\max}(\mathbf{E})$ are initially proportional to the dimensions $d$. This contributes to the observed imbalance in their singular values at the start of training.

# C    More experiments

## C.1    Analysis of singular values of embedding layer

Previous studies (e.g., [25], [10], [18], [26]) have explored the gap between SGD and Adam in optimizing Transformer models, but the specific role of embeddings and their bilinearity with downstream weights remains underexplored. For example, [25] attributes SGD's suboptimal performance to the heavy-tailed distribution of stochastic gradient noise. This observation aligns with our findings regarding the randomness in embedding updates for low-$p$ tokens.

On the other hand, [10] argues that gradient noise alone cannot explain Adam's superiority. Their experiments demonstrate that, even with full-batch training to eliminate stochastic noise, SGD underperforms compared to Adam. They suggest that the sign of the gradient might be a more reliable descent direction than its magnitude, and since Adam optimally balances both, it outperforms SGD, particularly in small-batch settings.

Furthermore, [26] provides a novel explanation for Adam's advantage over SGD in Transformers by analyzing the blockwise Hessian spectrum, introducing the concept of "block heterogeneity." This refers to significant variations in the Hessian spectra across parameter blocks, a phenomenon observed in Transformers but not in CNNs. However, the underlying source of this heterogeneity is not explicitly discussed. We hypothesize that this stems from the bilinear nature of weights, particularly in the embedding and attention mechanisms. To support this hypothesis, we analyze the Hessian of embedding weights compared to other weight below.

To analyze the curvature of the loss landscape, we compute the maximum eigenvalue of the Hessian matrix using the power method with Hessian-vector products (HVPs). This approach avoids explicitly constructing the Hessian, making it computationally efficient for large-scale systems.

The power method iteratively approximates the maximum eigenvalue of the Hessian $\mathbf{H}$ as follows:

1. Initialize a random vector $\mathbf{v}_0$ with the same dimensionality as the parameters $[\mathbf{E}, \mathbf{W}]$.

2. Compute the Hessian-vector product $\mathbf{H}\mathbf{v}_k$ using automatic differentiation:
$$\mathbf{H}\mathbf{v}_k = \nabla_{\boldsymbol{\theta}} \left( \nabla_{\boldsymbol{\theta}} \mathcal{L} \cdot \mathbf{v}_k \right),$$
where $\boldsymbol{\theta} = [\mathbf{E}, \mathbf{W}]$.

3. Normalize the vector and update the eigenvalue estimate:
$$\mathbf{v}_{k+1} = \frac{\mathbf{H}\mathbf{v}_k}{\|\mathbf{H}\mathbf{v}_k\|}, \quad \sigma_{\max} \approx \mathbf{v}_k^\top \mathbf{H}\mathbf{v}_k.$$

Figure 9 shows the maximum eigenvalues of the Hessian with respect to $\mathbf{E}$ and $\mathbf{W}$ during training. The results highlight distinct curvature properties for $\mathbf{E}$ and $\mathbf{W}$, reflecting their roles in the bilinear interaction.

Extending these insights to attention mechanisms highlights further challenges in bilinear optimization and demonstrates how adaptive learning rates (e.g., Adam) help escape saddle points. This suggests a deeper connection between the bilinearity of weight interactions and the optimization challenges unique to Transformer models.

## C.2    Rank Evolution and Implicit Regularization

Recent work has shown that weight decay in bilinear models (e.g., $\mathbf{Z} = \mathbf{E}\mathbf{W}$) implicitly regularizes the nuclear norm of the product matrix, promoting low-rank solutions and improved generalization

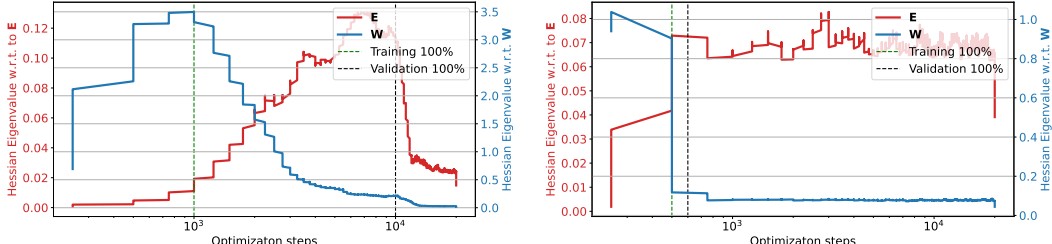

Figure 9: Maximum eigenvalues of the Hessian with respect to embedding weights (**E**) and downstream weights (**W**) during training. The left plot corresponds to the Adam optimizer, while the right plot uses Adam_lr optimizer (ours). With Adam (left), the eigenvalues for **E** are significantly smaller than those for **W**, reflecting differences in dimensionality and update frequency. In contrast, with Adam_lr (right), the eigenvalues of **W** are notably reduced and become closer to those of **E**, suggesting a more balanced optimization dynamic. Training accuracy reaches 100% when the eigenvalues of **W** begin to decrease, while validation accuracy improves as the eigenvalues of **E** decrease. This suggests that **W** drives early optimization progress, while **E** fine-tunes generalization. The Adam_lr optimizer (ours) appears to regularize **W**, leading to a more stable training process.

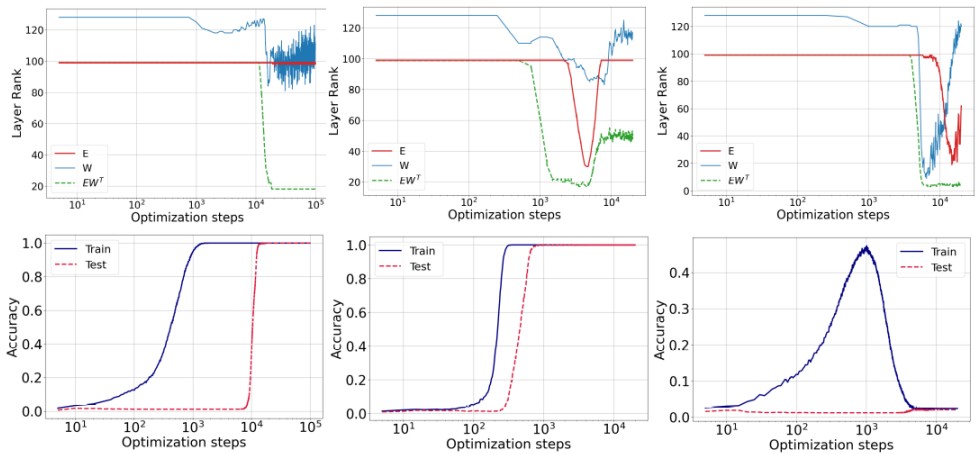

Figure 10: Rank evolution during training for three optimization setups: Adam (wd=0.001), Adam-LR (wd=0.001 with learning rate ratio), and Adam with stronger weight decay (wd=0.005). While all runs show decreasing rank(**EW**), only Adam-LR continues to adjust rank after generalization. This suggests that rank behavior alone does not fully explain grokking, and supports the need to analyze embedding-weight coupling dynamics.

[8]. This complements our focus on embedding dynamics, as both highlight the impact of bilinear coupling on optimization.

To explore this in our setup, we track the rank evolution of **E**, **W**, and the product **EW**. As shown in Figure 10, **W** exhibits three distinct phases: an early drop during training loss reduction, a plateau, and a final decline aligned with grokking. In contrast, **E**'s rank remains largely stable throughout.

Figure 10 compares three optimization setups: Adam (with weight decay 0.001), Adam-LR (our proposed variant with a learning rate ratio), and Adam with stronger weight decay (0.005). All configurations lead to a reduction in rank(**EW**), consistent with implicit nuclear norm regularization. However, only Adam-LR shows continued rank changes after generalization, suggesting that rank evolution alone does not capture the onset of grokking.

These findings reinforce that implicit regularization in bilinear systems depends not just on decay strength, but also on the interplay between initialization, update frequency, and curvature.

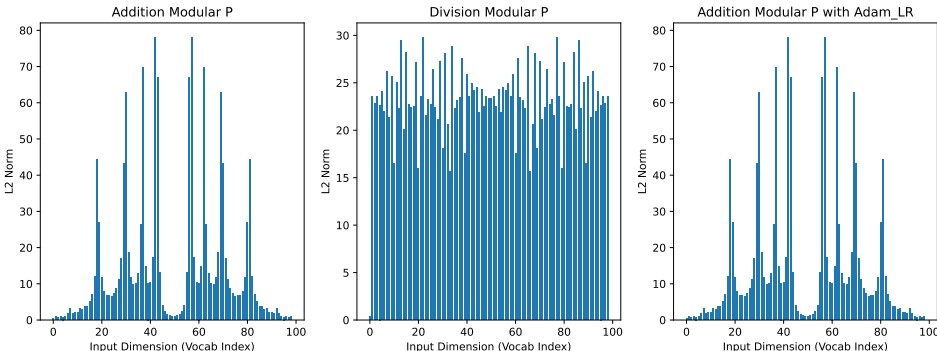

Figure 11: Discrete Fourier analysis of learned embedding representations across tasks. For each embedding matrix, we compute the DFT across the input dimension and the $\ell_2$-norm across the embedding dimension. Peaks indicate frequency localization that naturally aligns with the periodic structure of the task (e.g., modular addition), while tasks like modular division show more diffuse spectra.

## D Fourier Analysis of Embedding Representations

Fourier features offer a structured way to encode modular arithmetic directly into the input space. By encoding periodicity into the representation, such features can bypass the need for learned embeddings and mitigate challenges like sparse updates for rare tokens. However, this approach requires prior knowledge of the task's structure—e.g., periodicity—which may not apply in more complex tasks such as modular division or nonlinear compositions.

To investigate whether embedding layers naturally learn such structure, we analyze their frequency characteristics. Following the approach in [12], we apply the Discrete Fourier Transform (DFT) along the input dimension of the embedding matrix and compute the $\ell_2$-norm across the embedding dimension. We then plot the first $P/2$ components, leveraging the symmetry of the DFT.

The results for different tasks are shown in Figure 11. Clear frequency peaks indicate that the model internally captures task-specific periodic structure. Notably, such structure emerges even without explicit Fourier features, especially for modular addition and multiplication. However, in more complex tasks, such as modular division, this frequency localization diminishes—suggesting the limits of periodic encoding and the growing need for learned representations.

### D.1 Additional Datasets and Learning Rate Sensitivity

In addition to modular addition and division, we evaluate our methods on two further tasks: modular multiplication $(a \div b) \mod p$ and sum of squares $(a^2 + b^2) \mod p$. These tasks share the same architecture and tokenization as described in Section 5.

We emphasize that our experimental design is not centered on hyperparameter optimization. While aggressive tuning of learning rates and batch sizes can suppress or delay grokking, our goal is to study it where it naturally occurs. To that end, we identify configurations where grokking persists and focus our analysis there. This approach aligns with prior work on mechanistic understanding of grokking [9, 14], which likewise prioritize clarity of dynamics over benchmark performance. For illustration, Figures 13 and 14 show learning rate sensitivity on four datasets, confirming the robustness of our findings across reasonable settings (skewed distribution of embedding update delay the generalization).

### Compute Resources

All experiments were conducted using an NVIDIA A6000 GPU. Training runs were performed using PyTorch, with each configuration fitting comfortably within the GPU's 48 GB memory. No distributed training or multi-GPU setups were used.

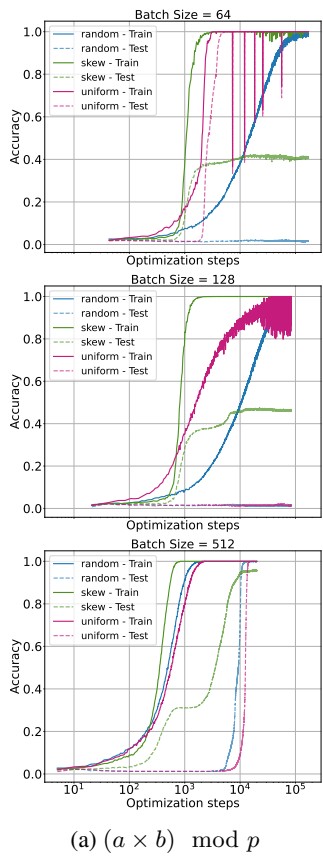

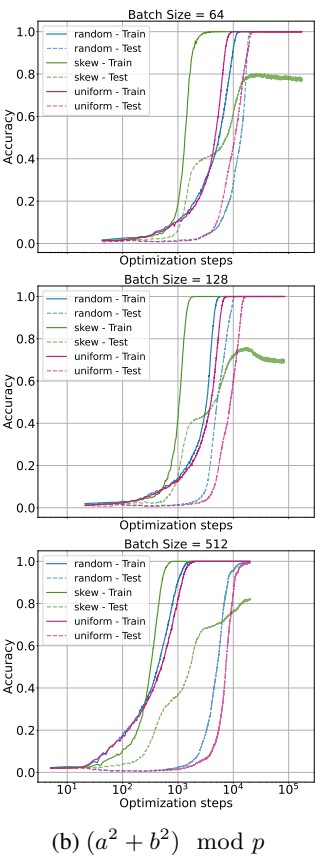

(a) $(a \times b) \mod p$

(b) $(a^2 + b^2) \mod p$

Figure 12: Sampling strategy comparison for multiplication and sum-of-squares tasks ($lr = 0.001$). Larger batch sizes narrow the performance gap, but skewed sampling still harms generalization.


Figure 13: Training and validation accuracies for the modular multiplication dataset for learning rate 0.01 across batch sizes (256, 512, 1024).

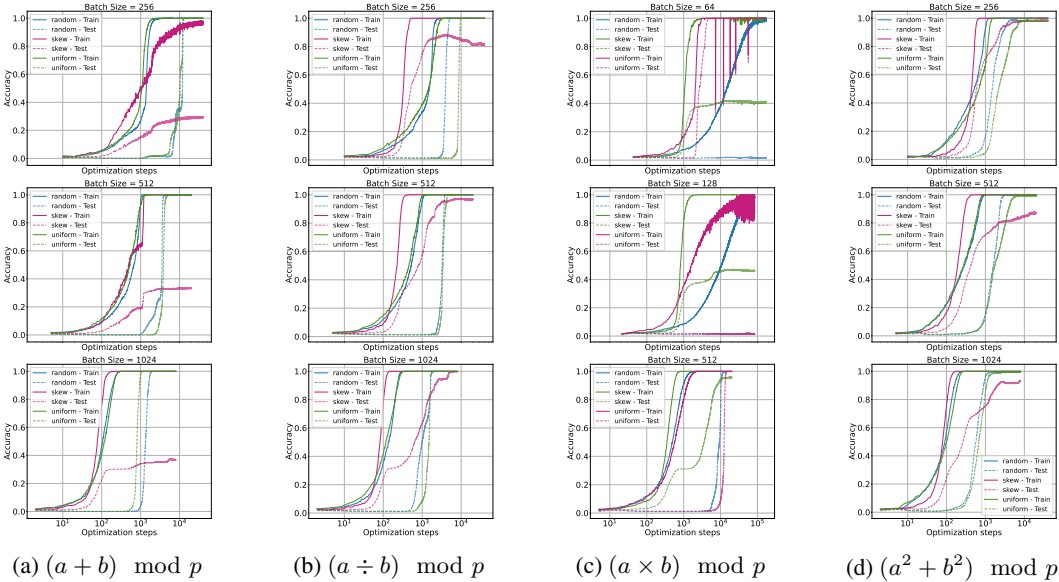

Figure 14: Training and validation accuracies for the modular multiplication dataset for learning rate 0.005 across batch sizes (256, 512, 1024).

acknowledge that the true answer is often more nuanced, so please just use your best judgment and write a justification to elaborate. All supporting evidence can appear either in the main paper or the supplemental material, provided in appendix. If you answer [Yes] to a question, in the justification please point to the section(s) where related material for the question can be found.

IMPORTANT, please:

- **Delete this instruction block, but keep the section heading "NeurIPS Paper Checklist",**
- **Keep the checklist subsection headings, questions/answers and guidelines below.**
- **Do not modify the questions and only use the provided macros for your answers**.

