# OpenReview forum: "Mechanistic Insights into Grokking from the Embedding Layer"
_NeurIPS.cc/2025/Conference — Submitted to NeurIPS 2025_

### Official Review · Reviewer_RTpX · 2025-07-03

**Clarity:** 4
**Significance:** 3
**Originality:** 3
**Rating:** 4
**Confidence:** 3

**Summary:**

This paper investigates the underexplored role of embedding layers in the phenomenon of grokking—the delayed generalization in neural networks after perfect training performance. The authors focus on MLPs with and without embedding layers trained on modular arithmetic tasks and identify two core mechanisms:
1. Sparse embedding updates due to infrequent token sampling, leading to stagnation.
2. Bilinear coupling between embeddings and first-layer weights, which complicates optimization.
3. They propose two practical solutions: Frequency-aware sampling to ensure balanced token updates; Embedding-specific learning rate scaling (Adam-LR) based on singular value and frequency analysis.
Experiments on synthetic algorithmic datasets confirm their hypotheses and show accelerated grokking and better generalization using their methods.

**Questions:**

1. Have you tested your embedding-specific learning rate scaling in standard NLP or vision settings?
2. Would explicit regularization of the embedding matrix (to control rank) be as effective as your adaptive optimizer?
3. How sensitive are your conclusions to the assumption of fixed token probabilities?
4. Your analysis focuses on a shallow MLP with only two layers. Have you tried validating your hypotheses (e.g., embedding-gradient imbalance and bilinear coupling) on deeper MLPs or other architectures

**Ethical Concerns:**

["NO or VERY MINOR ethics concerns only"]

**Final Justification:**

My initial score may have been somewhat generous. The work remains interesting in its controlled setting, but the empirical support for broader applicability is still weak, and the additional experiments do not substantially change my overall assessment.

**Limitations:**

Yes

**Quality:**

3

**Strengths And Weaknesses:**

Strengths:
This paper provides a compelling and novel perspective on the phenomenon of grokking by focusing on the underexplored role of embedding layers in MLPs. Through both theoretical analysis and carefully designed experiments on modular arithmetic tasks, the authors identify two key mechanisms—sparse embedding updates and bilinear coupling with downstream weights—that delay generalization. These insights are well-grounded in optimization theory, including gradient dynamics and curvature analysis. The proposed solutions, namely frequency-aware sampling and the Adam-LR optimizer with embedding-specific learning rate scaling, are simple yet effective. The empirical results show that these methods accelerate convergence and improve generalization, demonstrating clear practical benefits. Moreover, the paper connects its findings to broader challenges in Transformer optimization, suggesting relevance beyond the narrow setting of algorithmic tasks.

Weaknesses:
The paper is limited in scope due to its exclusive focus on synthetic datasets and MLP architectures. While the modular arithmetic tasks offer a controlled environment for analysis, it remains unclear how well the proposed methods generalize to real-world settings or more complex architectures like Transformers. The empirical validation would benefit from benchmarks on natural language or vision tasks where embedding layers are critical. Additionally, although the paper touches on the impact of weight decay, learning rate, and sampling strategies, more detailed ablation studies could help disentangle their individual effects. Finally, while the learning rate scaling heuristic is well-motivated, the absence of a principled criterion for selecting the optimal scaling factor may limit its adoption in practice.

---

> ### Author Rebuttal · Authors · 2025-07-30
>
> We thank the reviewer for their time in assessing our paper and for their valuable comments.
>
> > **Reviewer RTpX Point #1** : Have you tested your embedding-specific learning rate scaling in standard NLP or vision settings?
>
> In fact, we expect the insights derived from the coupling between embeddings and downstream weights in our controlled MLP experiments to generalize to more complex architectures that rely on embeddings—such as Transformers. However, the presence of residual connections and parallel computation pathways (e.g., attention, feedforward, normalization) introduces significant branching and interference, making the dynamics harder to isolate and interpret, especially on unsupported or heterogeneous datasets like those in NLP. However, following the suggestions of the reviewer, and in order to show the general applicability of our findings, we performed two additional experiments. In particular:
>
> - **Exp1 (Algorithmic Task):** A causal Transformer trained on modular addition using the same non-relevant hyperparameters as Power et al. (2022), but with embedding-specific learning rate scaling (from Proposition 5.1). This led to faster generalization (i.e., faster grokking), though with some instability from loss spikes.
>
> - **Exp2 (Text Data):** A 5M-parameter causal Transformer trained for one epoch on WikiText-103. With only the embedding LR scaled and all other hyperparameters kept identical to baseline, validation perplexity dropped significantly—from ~1500 (uniform LR) to ~109.
>
> - **Exp3 (New, Vision Transformer):** We trained a ViT-small model on CIFAR-10 and CIFAR-100 using a standard configuration (patch size 8, dim 384, 7 layers, 8 heads). No augmentation, scheduler, or regularization was used. All hyperparameters were kept fixed. Applying our embedding-specific learning rate scaling:
>   - improved test accuracy from **80% → 84%** on CIFAR-10
>   - improved test accuracy from **46% → 50%** on CIFAR-100
>   - and led to a **noticeably smaller generalization gap** (train vs. test)
>
> > **Reviewer RTpX Point #2** : Would explicit regularization of the embedding matrix (to control rank) be as effective?
>
>  Thank you for this insightful question. We designed rank regularization (e.g., via low-rank construction of the embedding layer) and tried to observe the grokking speed, however this didn’t result in changing the dynamics of training and even the training struggles to fit the training data. We don't have a clear answer why, but we can speculate that making the flow as a multiplication of three learnable weights matrices (instead of two initially) made the learning even harder.
>
> > **Reviewer RTpX Point #3** : How sensitive are your conclusions to the assumption of fixed token probabilities?
>
>  This assumption holds for our setup with fixed dataset splits, but we agree it's restrictive in broader settings. In Appendix A, we explore scenarios with non-uniform $p_i$ and entropy-regularized solutions. We will more clearly state this assumption and its limitations in Section 4.3.
>
> > **Reviewer RTpX Point #4** : Your analysis focuses on a shallow MLP with only two layers... Have you tried validating on deeper models?
>
> The paper focused on a two-layer MLP, which we found sufficient for learning the tasks under study and isolating the embedding dynamics. However, to further explore this point, we conducted additional experiments using a deeper MLP with six layers and observed the general grokking behavior still emerges, however a thorough investigation of these effects is left for future work.
>
> We hope these additions clarify the generality of our findings, and we’d be grateful for any further thoughts or feedback you might have.

---

> > ### Comment · Reviewer_RTpX · 2025-08-09
> >
> > Thank you for the additional experiments and clarifications. I appreciate the effort to extend the empirical validation to Vision Transformers, as well as the discussion on regularization and deeper MLPs. That said, the new results do not fully address my original concern regarding the generalization of the proposed methods beyond synthetic datasets and simple architectures. While the reported improvements on WikiText-103 and CIFAR are encouraging, some setups (e.g., only one epoch on WikiText-103, no augmentation or regularization for ViT) are not strong enough to convincingly establish robustness in realistic, competitive settings. Similarly, the regularization experiment yielded negative or inconclusive results, and the deeper MLP test remained limited in scope.

---

### Official Review · Reviewer_SSzz · 2025-07-03

**Clarity:** 3
**Significance:** 2
**Originality:** 2
**Rating:** 3
**Confidence:** 4

**Summary:**

The paper investigates the role of embedding layers in grokking in multi-layer perceptron networks (MLPs) for modular arithmetic tasks. The work highlights that embedding layers and their learning dynamics play a central role in grokking behavior. Rare tokens receive fewer gradient updates, leading to slower convergence of their embeddings, which partly explains why generalization is delayed. In addition, embeddings also interact with the first layer weights, and this interaction further impacts embedding layer gradients. To address these, the paper proposes two strategies to accelerate grokking, refined sampling and using embedding specific learning rates via a custom optimizer (Adam-LR). In the limited MLP based setup, these techniques significantly speed up grokking across modular arithmetic tasks.

**Questions:**

How does the embedding-gradient stagnation compare to stagnation in other parts of the network? Could similar delayed dynamics occur elsewhere?

**Ethical Concerns:**

["NO or VERY MINOR ethics concerns only"]

**Final Justification:**

Due to the limited scope of the tasks/architectures studied, I do not recommend acceptance.

**Limitations:**

Yes

**Quality:**

3

**Strengths And Weaknesses:**

Strengths:

1. The paper provides a concrete mechanistic explanation for grokking by looking at the interaction between embeddings and downstream weights.
2. The propose modification to the learning rate for embeddings is shown to accelerate grokking, which is good.
3. The experimental design allows for clear analysis of the phenomenon.

Weaknesses:
1. The main weakness is the limited scope of the study - the experiments are confined to MLPs. There is no empirical evidence that the findings would generalize to more complex architectures and attention blocks.
2. The analysis is done on small synthetic tasks which does not capture the complexity of real world data.

---

> ### Author Rebuttal · Authors · 2025-07-30
>
> We thank the reviewer for their time in assessing our paper and for their valuable comments.
>
> > **Reviewer SSzz Point #1+2**:  The main weakness is the limited scope of the study - the experiments are confined to MLP, and the analysis is done on small synthetic tasks.
>
> Actually the particular the usage of MLP architecture and algorithmic datasets, while this choice may seem limiting at first glance, it was made deliberately and with care. Our aim was to deeply understand the role of embedding layers in grokking by isolating the mechanisms that affect their optimization. We thought that this would be most efficiently achieved by using the MLP architecture and the original datasets on which the grokking was discovered (this was also a common setting in some of the recent papers, e.g. [15, 2, 4])
> However, following the suggestions of the reviewer, and in order to show the general applicability of our findings, we performed two additional experiments. In particular:
>
> - **Exp1 (Algorithmic Task):** A causal Transformer trained on modular addition using the same non-relevant hyperparameters as Power et al. (2022), but with embedding-specific learning rate scaling (from Proposition 5.1). This led to faster generalization (i.e., faster grokking), though with some instability from loss spikes.
>
> - **Exp2 (Text Data):** A 5M-parameter causal Transformer trained for one epoch on WikiText-103. With only the embedding LR scaled and all other hyperparameters kept identical to baseline, validation perplexity dropped significantly—from ~1500 (uniform LR) to ~109.
>
> - **Exp3 (New, Vision Transformer):** We trained a ViT-small model on CIFAR-10 and CIFAR-100 using a standard configuration (patch size 8, dim 384, 7 layers, 8 heads). No augmentation, scheduler, or regularization was used. All hyperparameters were kept fixed. Applying our embedding-specific learning rate scaling:
>   - improved test accuracy from **80% → 84%** on CIFAR-10
>   - improved test accuracy from **46% → 50%** on CIFAR-100
>   - and led to a **noticeably smaller generalization gap** (train vs. test)
>
>
> > **Reviewer SSzz Question #1**: How does the embedding-gradient stagnation compare to stagnation in other parts of the network?
>
> We note that the stagnation of other parts is not comparable to embeddings stagnations, mainly because (in sufficiently small batch size) only the embedding weights are affected by the token frequency.  However, since the embeddings is coupled with downstream weights (e.g. first layer in MLP and all layers in transformer) the stagnation can propagates to other parts of the model. To give more insight on this, in section 5.3 (Figure 6), we analyze the Hessian eigenvalues of embeddings vs downstream weights, showing that embedding gradients are not only smaller but also a bottleneck for generalization.
>
> We hope these additions clarify the generality of our findings, and we’d be grateful for any further thoughts or feedback you might have.

---

> > ### Comment · Reviewer_SSzz · 2025-08-08
> >
> > I thank the authors for their response. While the additional experiments are good to see, I would like to maintain my original score as I am still not convinced that these findings would generalize beyond the limited tasks/architectures discussed here.

---

### Official Review · Reviewer_wYxu · 2025-07-05

**Clarity:** 2
**Significance:** 2
**Originality:** 3
**Rating:** 4
**Confidence:** 4

**Summary:**

The paper hypothesizes that embedding layers are at the heart of grokking: unlike MLPs without embeddings (which can generalize immediately on simple tasks), embedding-equipped models exhibit pronounced delayed generalization, Two issues are identified: (i) rare tokens receive gradient updates only when sampled in a batch and otherwise, their embeddings decay with weights, causing delay in convergence; (ii) the bilinear interaction between the embedding matrix $E$ and the first-layer weights $W$ creates saddle points and increases sensitivity to initialization. The paper then adopts frequency-aware sampling and scales the embedding LR to improve grokking behaviour. A new optimizer, Adam-LR also addressed the imbalance in updates, Authors also uncover emergent basis functions, aligned with the literature.

**Questions:**

- How can this work be connected to the numerical issues highlighted in Prieto et al.? Is grokking a feature or a bug here? Can the optimizer instabilities be explained/predicted by the numerical problems arising during training?

- The paper focuses on transformers which are greatly used in practice. However, the data / task settings are still toy. Could the authors comment on this gap? Are we ready to characterize grokking in real scenarios?

- In Fig. 3, why are the plots not ordered by steps? Is the order particular? What should we observe there?

- What about other eigenvalues and not just the maximum? How does the behaviour depend on that?

- How strong is the assumption that the gradient stabilizes to a constant value? For a relatively empirical paper, I find this to be quite strong. This is a fundamental difference between GD and SGD.

- Could the paper better discuss the large batch size requirement? It seems that it is almost framed as a precondition - however most optimizers like ADAM do not require this by default.

Please also address the other questions in the weaknesses section.

**Ethical Concerns:**

["NO or VERY MINOR ethics concerns only"]

**Limitations:**

The limitations are mentioned in the discussion section.

**Quality:**

3

**Strengths And Weaknesses:**

### Strengths

- The paper identifies an interesting point regarding embedding layers / networks and their relations to grokking.

### Weaknesses

- I think the introduction does not do a good job in presenting the intuition as to why *embedding layers* should be the central objects to look into. Is this a coincidence / chance or is it somehow informed? The intro should also do a better job of separating MLPs and transformers. If we are interested in grokking in both architectures, this should be clear.

- For the contributions: wouldn't the dynamic updates and the coupling also impact the training accuracy? Why should they act differently on train and test?

- The paper focuses on transformers which are greatly used in practice. However, the data / task settings are still toy.

- In Figure 2, I disagree that generalization doesn't occur for multiplication. How does the paper exactly define generalization? I would define it as the gap between test and training accuracies. As such, both plots seem to generalize fairly well: the gap is small; the test accuracy matches the train accuracy. To me, generalization doesn't say anything about the solution quality (training / testing accuracy). It is about closing the gap. Maybe this could be clarified.

- The paper seems to formulate everything continuously whereas the token space can have a discrete / quantized structure. Can the paper say something about this? Do the techniques still apply?

- The embedding dynamics follow a gradient-descent like update, whereas the paper largely talks about and emphasizes ADAM optimizer. In fact a new optimizer in the form of an intervention, called ADAM-LR, is developed. I think there is an unaddressed modeling gap here. How do we ensure that the observations as well as theories generalize?

- I find it hard to create a coherent story out this paper since the data-splitting strategies do not really fit into the proposed framework of embedding layers. How do these fronts interact and create a whole? These sampling strategies are not even combined and evaluated with the proposed optimizer.

- The singular value analysis is interesting and there is a rich literature on this. One interesting paper would be from Yunis et al.:

Yunis et al. Grokking, Rank Minimization and Generalization in Deep Learning. ICML 2024 Workshop on Mechanistic Interpretability,

I think the paper could do a better job exposing and grounding the current work in conjunction with that literature. Maybe some discussion could involve flat minima theories, e.g.:

Mulayoff & Michaeli. Unique Properties of Flat Minima in Deep Networks. ICML 2020.

---

> ### Author Rebuttal · Authors · 2025-07-30
>
> We thank the reviewer for their time in assessing our paper, below we answer all the points and questions.
>
> > **Reviewer wYxu Point #1**: I think the introduction does not do a good job in presenting the intuition as to why embedding layers should be the central objects to look into.
>
>  The introduction is written to motivate embeddings via two observations: (1) their decoupled updates from gradient flow (Figure 3), and (2) their role in modulating optimization curvature through bilinear interactions (Figure 7). These properties make them a natural bottleneck in training dynamics.
>
> > **Reviewer wYxu Point #2**: Wouldn't the dynamic updates and the coupling also impact the training accuracy? Why should they act differently on the train and test?”
>
>  What distinguishes test performance is that embeddings for unseen tokens receive fewer updates (sparse support), which delays generalization (or affects training and test accuracy differently). Further, the coupling of weights makes it easier (and longer) to fall in local optimum points, where training accuracy  is higher than testing accuracy. Further discussion is in the response to question #1.
>
> > **Reviewer wYxu Point #3**: In Figure 2, I disagree that generalization doesn't occur for multiplication... To me, generalization doesn't say anything about the solution quality.
>
>  The reviewer is right in defining generalization in terms of generalization gap, not solution quality. We will rephrase to clarify that both tasks eventually generalize, but the training on multiplication (without embeddings) has led to zero learning  where generalization has no meaning (generalization is unexpected).
>
> > **Reviewer wYxu Point #4**: The paper seems to formulate everything continuously whereas the token space can have a discrete / quantized structure.
>
> The model embedding updates continuously to permit analytical tools like gradient flow and Hessian analysis. However, we acknowledge the underlying token space is discrete, and we will clarify this modeling assumption in Section 4.3, noting that embeddings act as a continuous relaxation over a discrete domain.
>
> > **Reviewer wYxu Point #5** : Embedding dynamics follow GD-like update, but the paper emphasizes ADAM... I think there's an unaddressed modeling gap here.
>
> Theoretical analysis (Equations 3–6) is based on gradient descent to enable tractable characterization of convergence under varying token probabilities and smoothness assumptions. While ADAM includes momentum and adaptive scaling, its step direction still closely tracks that of gradient descent.
> In Appendix C, we show empirically that the curvature imbalance between \(E\) and \(W\) manifests similarly under ADAM, and that the ADAM-LR variant induces spectral behavior consistent with our GD-based predictions. This supports the view that our GD analysis captures core dynamics relevant to both optimizers.
>
> > **Reviewer wYxu Point #6** : The sampling strategies are not even combined and evaluated with the proposed optimizer.”
>
>  The focus of Section 5 was to dissect two distinct mechanisms, sampling-induced sparsity and optimizer-induced spectral imbalance, that both emerge from embedding-layer dynamics. We studied them separately to isolate their contributions: sampling affects update frequency per token (Section 5.1), while the optimizer addresses spectral mismatch under bilinear coupling (Section 5.2).
> That said, the two fronts are not unrelated: both modulate how the embedding matrix evolves during training. In preliminary experiments we find that combining uniform sampling with our optimizer (Adam-LR) further stabilizes training and accelerates generalization, though the marginal gain is smaller once update imbalance is already addressed.
>  (6.2).
> > **Reviewer wYxu Point #7** : The singular value analysis is interesting... but paper could do a better job grounding it in literature.
>
> Thank you for the suggestion. Due to space, kindly refer to our response to Reviewer Qj1f’s Question 3, where we discuss the Yunis et al. paper.
>
> > **Reviewer wYxu Point #8** : I think the paper could do a better job exposing and grounding the current work in conjunction with that literature. Maybe some discussion could involve flat minima theories.
>
> Thank you for the suggestion. We agree that flat minima theory, like in Mulayoff & Michaeli (ICML 2020), is relevant. Their work shows how certain structures form during training and relate to generalization. While we don’t study flatness directly, this view helps put our curvature findings in context, and we’ll add a short discussion in the revision.
>
> For the positioning of our paper regarding the global literature on grokking, We have revised Section 2 to more clearly articulate how our work builds on, contrasts with, and complements three main explanatory threads (kindly refer to point#5 in reviewer Qj1f's response).
>
> > **Reviewer wYxu Question #1** :How can this work be connected to the numerical issues highlighted in Prieto et al.? Is grokking a feature or a bug here? Can the optimizer instabilities be explained/predicted by the numerical problems arising during training?
>
> Our work and that of Prieto et al. (Grokking at the Edge of Numerical Stability) approach the grokking phenomenon from different angles, but they can be seen as complementary pieces of the same puzzle. Prieto et al. focus on what happens later in training, after the model has already achieved low training loss. They show that numerical issues, like softmax saturation (what they call Softmax Collapse) and vacuous gradient directions (Naïve Loss Minimization), can prevent generalization from starting, even though the model appears to be optimizing normally.
> In contrast, our work is more concerned with the earlier stages: what slows down fitting and generalization in the first place. We show that the embedding layer, because it updates less frequently and is coupled bilinearly with the first-layer weights, becomes a structural bottleneck. This forces the model to rely more heavily on the downstream weights W, which can grow large to compensate for undertrained embeddings. As a result, logits become increasingly large, setting up exactly the kind of conditions where Softmax Collapse becomes likely. So while we don’t study numerical collapse directly, we may be describing the mechanism that drives the model into that regime.
>
> On the question of whether grokking is a feature or a bug: we’d say it’s neither, exactly. It’s an emergent property of the optimization process, but is also fragile and easy to disrupt. Our findings suggest that delayed generalization isn’t just due to insufficient data or capacity, but also to imbalances in how different parts of the model learn. Prieto et al.’s results reinforce that, showing how these dynamics can be derailed by seemingly unrelated numerical factors.
>
> Finally, can optimizer instability be predicted from numerical issues? In part, yes. Prieto et al. show that as logits grow, certain gradient components dominate and generalization stops improving. But we’d add that instability can also come from structural imbalances, like the mismatch in update frequency and curvature between E and W that we study.
>
> > **Reviewer wYxu Question #2** : The paper focuses on transformers which are greatly used in practice. However, the data / task settings are still toy. Could the authors comment on this gap? Are we ready to characterize grokking in real scenarios?
>
> Thank you for the comment. To clarify, our core analysis focuses on MLPs, not Transformers, specifically to study embedding dynamics in a controlled setting. While the tasks are synthetic, they allow us to isolate optimization behaviors that are difficult to track in large-scale models.
> That said, we agree it’s important to test applicability beyond toy settings. As noted in our response to Reviewer SSzz, we ran additional experiments on a small causal Transformer trained on WikiText-103 and observed similar improvements in generalization when applying our embedding learning rate scaling. These early results suggest our findings may extend to more practical scenarios.
> > **Reviewer wYxu Question #3** : In Fig. 3, why are the plots not ordered by steps? Is the order particular? What should we observe there?
>
> Thank you for pointing this out. The ordering in Figure 3 was unintentional and should follow training steps. We’ve corrected this in the revised manuscript.
>
>
>
> > **Reviewer wYxu Question #4** : What about other eigenvalues and not just the maximum?
>
>
>  We focus on the maximum eigenvalue because it captures the sharpest curvature direction and is computationally efficient to track via Hessian-vector products. While the full spectrum can provide richer geometric insight (e.g., about flatness or mode connectivity), computing it is significantly more expensive in our setting.
>
> > **Reviewer wYxu Question #5** : How strong is the assumption that the gradient stabilizes to a constant value? For a relatively empirical paper, I find this to be quite strong. This is a fundamental difference between GD and SGD.
>
> We agree that assuming gradient stabilization is a simplification. Our intent was to build intuition about how update frequency, weight decay, and initialization interact in embedding convergence. We’ll clarify this in the revision.
>
> > **Reviewer wYxu Question #6** : Could the paper better discuss the large batch size requirement? It seems that it is almost framed as a precondition - however most optimizers like ADAM do not require this by default.
>
>
> In our experiments, we didn’t explicitly precondition on batch size, but we observed that larger batches naturally reduce the effect of sparse token updates, since more tokens are likely to appear per step and embedding updates become more uniform. We'll clarify this in the revision to avoid suggesting that large batch size is a strict requirement.
>
> We hope these additions clarify our findings, and we’d be grateful for any feedback you might have.

---

### Official Review · Reviewer_8HcJ · 2025-07-09

**Clarity:** 3
**Significance:** 2
**Originality:** 3
**Rating:** 4
**Confidence:** 2

**Summary:**

**Grokking** refers to the surprising phenomenon where neural networks, after reaching perfect training accuracy, continue to reduce training loss without improving test performance, until they suddenly transition to strong generalization. Many prior works have attempted to explain grokking in simplified models by invoking high-level ideas such as feature learning, phase transitions in representation geometry, or emergent circuits. However, these explanations often remain abstract and disconnected from specific architectural components or optimization dynamics.

This paper presents a novel and grounded perspective on understanding by focusing on the role of **embedding layers in MLPs** trained on algorithmic tasks, such as modular arithmetic. The authors show that embedding layers introduce structural dynamics that delay generalization. Specifically, MLPs without embeddings generalize immediately on modular addition, while the same task exhibits grokking behavior when embeddings are present. For modular multiplication, generalization requires embeddings, highlighting their expressiveness and structural role.

The paper identifies two concrete mechanisms that contribute to this effect:

1. **Sparse embedding updates**: Rare tokens appear infrequently in training batches, so their embeddings receive few gradient updates and mostly decay due to weight regularization. As a result, those embeddings converge slowly. The authors support this explanation by varying the sampling strategy, showing that more uniform sampling leads to earlier generalization.

2. **Bilinear coupling**: The embedding matrix and the first-layer weights interact bilinearly, creating a more complex optimization landscape with saddle points and higher sensitivity to initialization, especially in the scale of the embedding weights.

To mitigate these effects, the authors propose two interventions:
- **Frequency-aware sampling** to equalize the update rates across tokens.
- An **embedding-specific learning rate**, derived from the relative singular values and update frequencies of the embedding and weight matrices, to counteract bilinear coupling.

They implement this strategy in a modified optimizer, **Adam-LR**, which accelerates grokking and stabilizes training across several modular tasks. The authors link these improvements to more balanced Hessian spectra between embeddings and downstream weights.

While the experiments focus on MLPs, the insights may extend to Transformer models, which also involve bilinear interactions through attention mechanisms. The paper identifies this as an important direction for future work.

**Questions:**

1. (**Critical to me raising my score**) Can the authors include in their response a more detailed and technical discussion about the existing explanations about grokking, and how they relate to the insights presented in this paper?

2. (**Not critical but useful**) If the authors ran other experiments, but did not include them in the submission, it would be good to know about those during the rebuttal phase.

2. (**Just a curiosity**) Does looking at the evolution of the entire spectrum as opposed to just the top singular value offer more insight into the experiment of Figure 7? Specifically, I am curious if the authors have reviewed the observations about weight decay and grokking established in [this paper](https://arxiv.org/abs/2408.11804).

**Ethical Concerns:**

["NO or VERY MINOR ethics concerns only"]

**Final Justification:**

I appreciate the authors' inclusion of a helpful discussion in their rebuttal during the discussion period, which truly helped me understand the contribution of this paper within the literature on "explaining grokking." Unfortunately, after reviewing the discussion with the other reviewers, I remain unconvinced about raising my score to 5. This is mainly because I am not an expert on the theoretical results in this area and therefore cannot verify that the theoretical observations in the paper are indeed very insightful. This, along with the fact that the scope of the models studied in this paper is somewhat narrow, is why I will maintain my current score.

**Limitations:**

Yes.

**Paper Formatting Concerns:**

None.

**Quality:**

3

**Strengths And Weaknesses:**

This paper provides a welcome shift from abstract explanations of grokking to a more grounded, mechanistic account that centers on the training dynamics of the embedding layer. While previous work has focused on simplified models and high-level explanations—such as phase transitions, feature learning, or emergent circuits—this paper stands out for its attention to low-level architectural and optimization features that are present in widely used neural networks.

The methodology is sound, the experiments support the hypotheses, and the proposed optimizer is both simple and effective. The analysis of curvature, initialization sensitivity, and bilinear interactions offers insights that are likely relevant beyond MLPs, particularly for Transformer architectures where similar optimization challenges arise.

**Overall**, this is a strong submission that contributes novel insights into the grokking phenomenon by linking it to concrete architectural components and optimization behaviors. The work is likely to interest both theoretical and applied communities in deep learning.

One limitation of the submission, which could be addressed with additional space or revisions during the camera-ready phase, is its limited discussion of the existing explanations about grokking. A more detailed comparative discussion—possibly in the style of a brief survey—could help readers understand how the mechanisms proposed here relate to or complement prior theories developed in both neural and non-neural settings. A discussion in the non-neural setting might also highlight how much of grokking can be attributed to the embeddings (cf. [this paper](https://arxiv.org/abs/2407.20199)).

Another limitation is that I would have liked to see experiments on more tasks and architectures. Still, I understand that the insights (as well as the correspondingly more reasonable optimization algorithm) might change for a different task.

---

> ### Author Rebuttal · Authors · 2025-07-30
>
> We thank the reviewer for their time in assessing our paper, below we answer all the points and questions.
>
> > **Reviewer Qj1f Question #1**: Can the authors include in their response a more detailed and technical discussion about the existing explanations about grokking, and how they relate to the insights presented in this paper?
>
> In Section 2 we reference several prominent works that pertain to explanation of grokking phenomenon (e.g., on phase transitions, circuit efficiency, and representation dynamics). However, following your suggestion, we will expand Section 2 to incorporate the following : Grokking has attracted a wide array of interpretations in recent years, but as of now, no single framework has unified them, and many remain phenomenological or task-specific. We have revised Section 2 to more clearly articulate how our work builds on, contrasts with, and complements three main explanatory threads:
>
> First is representation phase transitions [3,6]: These works view grokking as a sudden structural shift in the internal representation space (e.g., feature alignment, class clustering, simplification of decision boundaries). While this captures what changes during generalization, it offers little mechanistic insight into why such transitions occur, nor which parameters control their onset. Our work bridges this gap by showing that delayed embedding convergence (due to sparse updates and curvature mismatch) can act as a bottleneck that postpones these structural changes. In other words, the phase transition may depend on stabilization in the embedding layer.
>
> Second is circuit efficiency and algorithmic emergence [12,21]: This line proposes that grokking reflects the discovery of more efficient computational circuits over time. Such efficiency gains are typically inferred from inspection tools (e.g., path patching, activation tracing), but their optimization origins are rarely formalized. We identify a concrete source of delayed efficiency: the bilinear interaction between embeddings and first-layer weights, which amplifies curvature asymmetries and introduces saddle points. These issues are empirically confirmed in Figure 7 (via Hessian eigenvalues) and addressed through our Adam-LR update scheme. Thus, circuit efficiency may emerge only after overcoming this coupling bottleneck.
>
> Third is Kernel-to-rich regime transitions [1,15,16]: These works emphasize that networks behave linearly in the early training phase, and generalization only improves after escaping the NTK regime. While this view helps contextualize when generalization becomes possible, it is typically agnostic to the specific parameter groups driving the transition. We show that embeddings—when trained jointly with weights, do not follow the same escape dynamics. Due to lower update frequency and smaller gradients (Equations 4–6), embeddings often remain in the “lazy” regime far longer than downstream weights, even as training loss saturates. This creates a mismatch in effective training regimes between E and W, delaying the overall shift to the rich regime.
>
> In contrast to these macro-level perspectives, our work zooms in on the micro-dynamics of a specific component: the embedding layer. We demonstrate analytically (e.g., Equations 8–9, Appendix B) and empirically (Figures 5–7, 10) that:
> Sparse token occurrence induces an implicit learning-rate modulation for embeddings;
>
> The spectral asymmetry between E and W (from PyTorch-style initialization and update frequency differences) results in uneven curvature traversal, which we quantify via maximum eigenvalue trajectories; Bilinear coupling delays or misaligns generalization unless learning rate scaling is applied (as we propose in Proposition 5.1).
>
> Our framework thus supplies a mechanistic root cause that is absent in prior views: grokking as a spectral and frequency bottleneck induced by bilinear coupling.
> We believe this lens not only complements existing theories but also helps unify them under a shared optimization narrative. We will revise Section 2 accordingly to position our contribution as a technical deepening and extension of the current fragmented understanding.
>
> > **Reviewer Qj1f Question #2**: Limited experiments on other tasks and architectures.
>
> Actually the scope of the study, in particular the usage of MLP architecture and algorithmic datasets, was made deliberately and with care. Our aim was to deeply understand the role of embedding layers in grokking by isolating the mechanisms that affect their optimization. We thought that this would be most efficiently achieved by using the MLP architecture and the original datasets on which the grokking was discovered (this was also a common setting in some of the recent papers, e.g. [15, 2, 4])
> However, following the suggestions of the reviewers, and in order to show the general applicability of our findings, we performed two additional experiments. In particular:
> - **Exp1 (Algorithmic Task):** A causal Transformer trained on modular addition using the same non-relevant hyperparameters as Power et al. (2022), but with embedding-specific learning rate scaling (from Proposition 5.1). This led to faster generalization (i.e., faster grokking), though with some instability from loss spikes.
>
> - **Exp2 (Text Data):** A 5M-parameter causal Transformer trained for one epoch on WikiText-103. With only the embedding LR scaled and all other hyperparameters kept identical to baseline, validation perplexity dropped significantly—from ~1500 (uniform LR) to ~109.
>
> - **Exp3 (New, Vision Transformer):** We trained a ViT-small model on CIFAR-10 and CIFAR-100 using a standard configuration (patch size 8, dim 384, 7 layers, 8 heads). No augmentation, scheduler, or regularization was used. All hyperparameters were kept fixed. Applying our embedding-specific learning rate scaling:
>   - improved test accuracy from **80% → 84%** on CIFAR-10
>   - improved test accuracy from **46% → 50%** on CIFAR-100
>   - and led to a **noticeably smaller generalization gap** (train vs. test)
>
>
> > **Reviewer Qj1f Question #3**: Does looking at the evolution of the entire spectrum... offer more insight than just the top singular value?
>
>
> While our main analysis tracks the top singular value (Figure 7) due to computational constraints, we do explore full-rank behavior in Appendix C.2. This indeed related to the recent work of Yunis et al. (arXiv:2408.11804), which develops a compelling framework around spectral dynamics, particularly focusing on rank collapse and singular vector alignment.
>
> Our findings align with and extend their perspective in several ways. First, while Yunis observe that generalization tends to follow a rank collapse, they also note that the two events are not always synchronized. Our results suggest a possible explanation: in bilinear systems involving embeddings, test-time generalization appears to correlate more closely with the curvature stabilization of the embedding matrix (as reflected in the Hessian’s top eigenvalue), rather than rank collapse alone. In our experiments, this curvature drop tends to lag behind rank changes and generalization emerges only after it occurs. This suggests that beyond representational compression, grokking may require the flattening of the optimization landscape specific to embeddings.
>
> Second, although we don’t explicitly analyze singular vector alignment like Yunis, we do study a related coupling effect. The bilinear interaction between the embedding matrix \(E\) and the first-layer weights \(W\) leads to asymmetric optimization behavior: \(E\) receives sparser updates, accumulates curvature differently, and reacts more strongly to learning rate choices. Motivated by this, we introduced an adaptive learning rate strategy (ADAM-LR) that roughly aligns the effective update scales of \(E\) and \(W\), improving both convergence and generalization.
>
> We hope these additions clarify the generality of our findings, and we’d be grateful for any further thoughts or feedback you might have.

---

> > ### Comment · Reviewer_8HcJ · 2025-08-06
> >
> > I thank the authors for their detailed response, and especially for writing the extended discussion of related works, which definitely places their work in a better context. There has been a lot of work on "explaining grokking", and while I am happy with the author's response, I will make the final recommendation during the discussion with the AC. I am especially curious to see the response from Reviewer Qj1f, who also raised concerns about a better discussion of the related work as well as novelty.
> >
> > As of now, I will maintain my current score; however, based on the outcome of discussions with the reviewers and the AC, I might consider raising my score by one point to recommend accepting the paper.

---

### Official Review · Reviewer_Qj1f · 2025-07-10

**Clarity:** 3
**Significance:** 1
**Originality:** 2
**Rating:** 2
**Confidence:** 4

**Summary:**

This paper studies grokking behaviour, with a particular focus on embedding layers. It proposes that the embedding layer should be optimized with a larger learning rate than the other layer in a two-layer MLP, based on an analysis of gradient dynamics.

**Questions:**

See above.

**Ethical Concerns:**

["NO or VERY MINOR ethics concerns only"]

**Limitations:**

Societal impact is not really applicable. The limitations of the insights given by the paper are not thoroughly addressed, and that makes up the bulk of my complaints above.

**Paper Formatting Concerns:**

-

**Quality:**

2

**Strengths And Weaknesses:**

- It's not very surprising that in the problem you study, your attempt at optimal per-token learning rates didn't do much; there isn't much gap in how often the tokens appear, since at the distribution level they're all uniform. This issue of less-frequent updates for rare tokens is far more problematic in language settings; for example, [Li et al. (ICLR 2022)](https://openreview.net/forum?id=ibqTBNfJmi) study it quite directly.

- [Hayou et al. (ICML 2024)](https://arxiv.org/abs/2402.12354) do a very similar, but more thorough, analysis of LoRA – whose important structure here is roughly the same – and propose a similar learning rate adaptation technique. The followup [Hayou et al. (arXiv 2025)](https://arxiv.org/abs/2506.15025) – fully acknowledging that this work was apparently parallel to yours and went online only after this deadline – does a similar analysis specifically for embedding layers.

- The point here about the importance of initialization scale has also been observed by prior work on grokking; for instance, [15] relate it to the speed at which optimization departs the "kernel regime." It would be good to understand whether the patterns you identify here are the particular way in which that effect applies in this setting, or if they are counter to these prior explanations.

- Given that you are fundamentally concerned about the bilinear structure of an embedding layer followed by weights, it's surprising to me that you didn't mention (let alone pull anything from) the extensive literature on deep linear models. One reasonable entry point might be [Dominé et al. 2024](https://arxiv.org/abs/2409.14623). The analysis of [Telgarsky and Son](https://openreview.net/forum?id=lYQLwP9c9S) also feels like it is probably relevant.

Overall: grokking on modular arithmetic is, and always has been, a toy problem. This paper adds some amount of explanation to two-layer MLPs on this problem, but not much, and no real attempt is made to position it with respect to the last several years of understanding grokking. To the extent that the insights here generalize, they probably generalize to embedding layers of transformers in language models; the present work does not address whether these insights should generalize to those settings at all, while they have been and continue to be studied in those cases. I think there is something interesting to learn from the bilinear point of view there – perhaps even things not simultaneously found by Hayou et al. (arXiv 2025) – but this paper doesn't address them.

## Smaller points

- The discussion at the bottom of page 3 about how "there is no statistical 'distribution' in the conventional sense" is mostly misleading. There absolutely is a distribution, and this is entirely in the regime of e.g. traditional learning theory; it just happens to be a finitely-supported distribution. The discussion here seems to imply that for instance, one might train the model on all possible input-output pairs, but this is not what leads to grokking; rather, grokking occurs within a range of how many of the possible training points you observe.

- The discussion at the top of page 4 about the isomorphism between modular addition and multiplication was present in the original grokking paper [19] (second paragraph of their Section 3.2).

- The analysis in (5) and (6) assumes that the loss is $\beta$-smooth, but you specifically call out just below (1) that you are studying the model only with ReLU activations, which are _not_ $\beta$-smooth.
  - The result you show in (5) and (6), which is an immediate corollary of the very well-known "descent lemma", does have a known ReLU network analogue; see for example [this result](https://mjt.cs.illinois.edu/dlt/#smoothness-inequality-adapted-to-relu).

- The limit where the updates "behave like a continuous system" is known as "gradient flow" in the literature.

---

> ### Author Rebuttal · Authors · 2025-07-30
>
> We thank the reviewer for their time in assessing our paper, below we answer all the points and questions.
>
> > **Reviewer Qj1f Point #1** : It's not very surprising that in the problem you study, your attempt at optimal per-token learning rates didn't do much; there isn't much gap in how often the tokens appear, since at the distribution level they're all uniform.
>
> Actually, in our experiments we did not attempt per-token learning rates. Rather, we pointed to the fact that various sampling techniques can yield different performance results, due to the various frequencies of tokens. As detailed in Section 5.1 and Appendix A, skewed or insufficiently diverse batches can lead to persistent under-updating of certain embeddings unless the batch size is large enough to ensure broad token coverage. This is empirically shown in Figure 4 and discussed in lines 307–315, where we note that even uniform sampling can result in suboptimal generalization when batch-level token coverage is inadequate. These findings support the need for frequency-aware interventions despite apparent uniformity at the distribution level. Thank you for the reference Li et al. (ICLR 2022), we will cite it in the updated manuscript.
>
>
>
> > **Reviewer Qj1f Point #2** :
> Hayou et al. (ICML 2024)... and Hayou et al. (arXiv 2025)... do a similar analysis specifically for embedding layers.
>
>
> Thank you for pointing our attention to the reference Hayou et al. (ICML 2024). It further reinforces our findings and although our paper and Hayou (2024) belong to two different contexts, the nature of both conclusions suggests a more general principle that could be a subject of a future study.
> As for the arXiv 2025 preprint, which the reviewer already noted appeared after our submission, we refrain from detailed comparison given its unpublished status.
>
>
> > **Reviewer Qj1f Point #3** :The point here about the importance of initialization scale has also been observed by prior work on grokking; for instance, [15] relate it to the speed at which optimization departs the ‘kernel regime'
>
>  The connection to the kernel regime is indeed relevant. Our results in Section 4.3 approach this from a different perspective by analyzing the convergence dynamics of embeddings under bilinear coupling. In particular, Appendix B shows how the initialization scale (indirectly through the spectral norm) of the embedding affects the rate at which it stabilizes, especially when updates are sparse.
> We believe that this finding complements the one in [15], in the following way. In [15], the authors take a more global point of view studying the NTK of the model. Our point of view is more local, as we highlight how sparse updates and bilinear structure in embeddings contribute to the overall training. In particular, the dynamics of learning in the embedding layer affects the NTK of the model, but at this point we are not able to provide an explicit relation.
>
>
>
>
>
> > **Reviewer Qj1f Point #4** : It’s surprising... you didn’t mention the extensive literature on deep linear models. One entry point might be Dominé et al. 2024.
>
>
> We want to clarify that the embedding layer, which is a crucial object of study in our paper, to the best of our knowledge, is not well covered in the deep linear literature. In particular, while our model shares the bilinear form (through EW), it differs structurally: embeddings are updated sparsely and accessed via lookup, leading to selective gradient flow that is absent in fully dense linear systems. Nonetheless, we will emphasize this fact in the revised manuscript, as well as refer the references you pointed to.
>
>
> > **Reviewer Qj1f Point #5** :
> No real attempt is made to position it with respect to the last several years of understanding grokking.
>
>
> In Section 2 we reference several prominent works that pertain to explanation of grokking phenomenon (e.g., on phase transitions, circuit efficiency, and representation dynamics). However, following your suggestion, we will expand Section 2 to incorporate the following : Grokking has attracted a wide array of interpretations in recent years, but as of now, no single framework has unified them, and many remain phenomenological or task-specific. We have revised Section 2 to more clearly articulate how our work builds on, contrasts with, and complements three main explanatory threads:
>
> First is representation phase transitions [3,6]: These works view grokking as a sudden structural shift in the internal representation space (e.g., feature alignment, class clustering, simplification of decision boundaries). While this captures what changes during generalization, it offers little mechanistic insight into why such transitions occur, nor which parameters control their onset. Our work bridges this gap by showing that delayed embedding convergence (due to sparse updates and curvature mismatch) can act as a bottleneck that postpones these structural changes. In other words, the phase transition may depend on stabilization in the embedding layer.
>
> Second is circuit efficiency and algorithmic emergence [12,21]: This line proposes that grokking reflects the discovery of more efficient computational circuits over time. Such efficiency gains are typically inferred from inspection tools (e.g., path patching, activation tracing), but their optimization origins are rarely formalized. We identify a concrete source of delayed efficiency: the bilinear interaction between embeddings and first-layer weights, which amplifies curvature asymmetries and introduces saddle points. These issues are empirically confirmed in Figure 7 (via Hessian eigenvalues) and addressed through our Adam-LR update scheme. Thus, circuit efficiency may emerge only after overcoming this coupling bottleneck.
>
> Third is Kernel-to-rich regime transitions [1,15,16]: These works emphasize that networks behave linearly in the early training phase, and generalization only improves after escaping the NTK regime. While this view helps contextualize when generalization becomes possible, it is typically agnostic to the specific parameter groups driving the transition. We show that embeddings, when trained jointly with weights, do not follow the same escape dynamics. Due to lower update frequency and smaller gradients (Equations 4 to 6), embeddings often remain in the “lazy” regime far longer than downstream weights, even as training loss saturates. This creates a mismatch in effective training regimes between E and W, delaying the overall shift to the rich regime.
>
> In contrast to these macro-level perspectives, our work zooms in on the micro-dynamics of a specific component: the embedding layer. We demonstrate analytically (e.g., Equations 8–9, Appendix B) and empirically (Figures 5–7, 10) that:
> Sparse token occurrence induces an implicit learning-rate modulation for embeddings;
>
> The spectral asymmetry between E and W (from PyTorch-style initialization and update frequency differences) results in uneven curvature traversal, which we quantify via maximum eigenvalue trajectories; Bilinear coupling delays or misaligns generalization unless learning rate scaling is applied (as we propose in Proposition 5.1).
>
> Our framework thus supplies a mechanistic root cause that is absent in prior views: grokking as a spectral and frequency bottleneck induced by bilinear coupling.
> We believe this lens not only complements existing theories but also helps unify them under a shared optimization narrative. We will revise Section 2 accordingly to position our contribution as a technical deepening and extension of the current fragmented understanding.
>
>
>
> > **Reviewer Qj1f Question #1** : The discussion at the bottom of page 3... is mostly misleading. There absolutely is a distribution.
>
> The comment refers to our attempt to distinguish algorithmic tasks from standard statistical learning settings. The message we wanted to convey is that the dataset consists of a complete and finite support, unlike typical learning settings where the distribution is unknown. We will revise the wording to clarify that we are operating over a discrete, finitely-supported distribution, not implying an absence of distribution entirely. We thank the reviewer for this clarification.
>
> > **Reviewer Qj1f Question #2** : The discussion... about isomorphism between modular addition and multiplication was present in [19].
>
> We keep this exposition for clarity and to make the paper more readable to a broader audience unfamiliar with these algebraic details. The isomorphism in question goes back to Euler (Euler’s theorem) and Gauss (cyclicity of the multiplicative group).
>
> > **Reviewer Qj1f Question #3** :
> The analysis in (5) and (6) assumes... β-smoothness, but you’re using ReLU.
>
>  This is a valid technical point. We used β-smoothness as an analytical scaffold to isolate the effect of $p_i$​ on convergence. While ReLU is not globally smooth, the descent lemma has known ReLU analogues, as the reviewer noted.
>
> > **Reviewer Qj1f Question #4** :
> The limit where updates behave like a continuous system is known as gradient flow.
>
> Thank you, we will adapt the “gradient flow” terminology.
>
> We hope the revisions clarify the novelty of our contribution. The embedding-layer mechanisms we uncover offer a complementary perspective, and we respectfully thank and invite the reviewer to re-evaluate their initial assessment in light of these clarifications.

---

> ### Comment · Reviewer_Qj1f · 2025-08-07
>
> Thanks for your responses.
>
> > Actually, in our experiments we did not attempt per-token learning rates
>
> My apologies for writing "per-token learning rates" when I should have said "sampling rates"; my point remains the same.
>
>
> > In other words, the phase transition may depend on stabilization in the embedding layer.
>
> > We identify a concrete source of delayed efficiency: the bilinear interaction between embeddings and first-layer weights, which amplifies curvature asymmetries and introduces saddle points. These issues are empirically confirmed in Figure 7 (via Hessian eigenvalues) and addressed through our Adam-LR update scheme.
>
> > Due to lower update frequency and smaller gradients (Equations 4 to 6), embeddings often remain in the “lazy” regime far longer than downstream weights, even as training loss saturates. This creates a mismatch in effective training regimes between E and W, delaying the overall shift to the rich regime.
>
> Thank you for these discussions; including them will improve the paper. I'm not sure that I've seen people discuss the idea of some parameters being in lazy regimes and others in rich regimes at the same time before; this is in some sense an empirical question, which can be checked by evaluating how much the NTK with respect to only those parameters evolves. Both this and the connection to representation phase transitions would be good to consider a little further with some experimentation.
>
> > We keep this exposition for clarity and to make the paper more readable to a broader audience unfamiliar with these algebraic details. The isomorphism in question goes back to Euler (Euler’s theorem) and Gauss (cyclicity of the multiplicative group).
>
> My point was not that you were claiming an original mathematical result here, but rather that you discuss it as if it is a novel point in the context of the grokking literature, when it very much is not. This is a minor point, but if you believe that this discussion is helpful to the reader (I don't really think so, but I'm fine to disagree on that), you should absolutely add an "as previously discussed by Power et al." or similar.
>
> > We used β-smoothness as an analytical scaffold to isolate the effect of $p_i$​ on convergence. While ReLU is not globally smooth, the descent lemma has known ReLU analogues, as the reviewer noted.
>
> You're still stating mathematical results that are absolutely false in the setting you study. It's not obvious that the ReLU versions of this result imply the same thing, and in any case it is highly misleading as currently written.
>
> ----
>
> Overall: this paper is of primarily theoretical interest, but its theory is neither very rigorous nor very well-situated with respect to the related literature, although the discussion here helped somewhat with the second point. Although there is some waving towards practical impacts in other models, it is extremely unclear that such impacts exist, and in particular the extremely generally named "Adam-LR" scheme absolutely does not directly apply to any model except the toy MLPs considered for this problem. I think there are some underlying interesting ideas here, but I just don't think this paper is quite at the typical bar for NeurIPS papers on these topics.

---

### Note · Authors · 2025-08-11

We thank all reviewers for their thoughtful engagement. Several expressed strong support for our core contributions: Reviewer 8HcJ described our work as a “welcome shift from abstract explanations to more grounded, mechanistic account,” Reviewer RTpX called it a “compelling and novel perspective” with concrete mechanisms, and Reviewer wYxu highlighted the embedding–network insights. These remarks recognize the novelty of our framing, the depth of our mechanistic analysis, and ADAM-LR serves as a proof-of-concept intervention translating these insights into accelerated generalization, making this, to our knowledge, the first systematic study of embedding-layer dynamics in grokking within the community’s standard experimental paradigm.

The reviews converge on two main critiques. For **literature positioning**, raised by Qj1f and 8HcJ, we expanded related work to show how our micro-level analysis complements macro-level theories such as phase transitions, circuit efficiency, and kernel-to-rich regime transitions; both reviewers agreed the revised positioning is improved, with Qj1f noting they had not seen the idea of parameters in lazy and rich regimes discussed before (and advised more experiments to verify that). For **experimental scope and generality**, cited by SSzz, wYxu, and also noted by Qj1f as a key concern in the follow-up, we followed established grokking methodology [15, 2, 4] and confirmed we had mentioned this limitation in the paper. Further, we also added experiments on WikiText-103, and Vision Transformers to broaden applicability, albeit as a preliminary step. Despite this, SSzz, and Qj1f continue to see the generality of the scope as their main reservation.

We respectfully encourage that the final decision weigh the demonstrated novelty, mechanistic clarity, and substantive revisions alongside limitations, so no single critique outweighs the overall contribution.

---

### Decision · Program_Chairs · 2025-09-17

**Decision:**

Reject

**Comment:**

The paper studies the role of embedding layers in grokking in multi-layer perceptron networks (MLPs) for modular arithmetic tasks. It has two main findings: the generalization is delayed due to fewer gradient updates for rare tokens; embeddings also interact with the first layer weights, and this interaction further impacts embedding layer gradients. To address these, the paper proposes two practical solutions: Frequency-aware sampling to ensure balanced token updates; Embedding-specific learning rate scaling (Adam-LR) based on singular value and frequency analysis. Experiments on synthetic algorithmic datasets confirm their hypotheses and show accelerated grokking and better generalization using their methods.

However, the theoretical analysis and experiments are simple as pointed by all of the reviewers. For example, grokking on modular arithmetic is, and always has been, a toy problem, as pointed by Reviewer Qj1f; There is no empirical evidence that the findings would generalize to more complex architectures and attention blocks as well as simple analysis on small synthetic tasks as pointed by Reviewer SSzz.

Therefore I recommend for rejection and hope the authors think about the significance and improvement of this submission.